



# Joint calibration of multi-scale hydrological data sets using probabilistic water balance data fusion: methodology and application to the irrigated Hindon River Basin, India

Roya Mourad[1], Gerrit Schoups[1], Vinnarasi Rajendran[2], Wim Bastiaanssen[1,3]

[1]Faculty of Civil Engineering and Geosciences, Delft University of Technology, Mekelweg 5, Delft, 2628 CD, the Netherlands
[2]Civil Engineering Department, Indian Institute of Technology Roorkee, Roorkee, Uttarakhand , India
[3]Hydrosat, Wageningen, Agro Business Park 10, 6708 PW, the Netherlands.

*Correspondence to*: Roya Mourad (r.mourad@tudelft.nl)

**Abstract.** Hydrological data sets have vast potential for water resource management applications; however, they are subject to uncertainties. In this paper, we develop and apply a monthly probabilistic water balance data fusion approach for automatic bias correction and noise filtering of multi-scale hydrological data. The approach first calibrates the independent data sets by linking them through the water balance, resulting in hydrologically consistent estimates of precipitation ($P$), evaporation ($E$), storage ($S$), irrigation canal water imports ($C$), and river discharge ($Q$) that jointly close the basin-scale water balance. Next, the basin-scale results are downscaled to the pixel-scale, to generate calibrated ensembles of gridded Precipitation ($P$) and Evaporation ($E$) that reflect the basin-wide water balance closure constraints. An application to the irrigated Hindon River basin in India illustrates that the approach generates physically reasonable estimates of all basin-scale variables, with average standard errors of 21 mm month[-1] for storage, 7 mm month[-1] for precipitation, 10 mm month[-1] for evaporation, 4 mm month[-1] for irrigation canal water imports, and 2 mm month[-1] for river discharge. Results show that updating the original independent data with water balance constraint information reduces uncertainties by inducing cross-correlations between them. In addition, the introduced approach yields (i) hydrologically consistent gridded $P$ and $E$ estimates that fuse information from prior (original) data across different land use elements and (ii) statistically consistent random errors that reflect the model's confidence about $P$ and $E$ estimates at each grid cell. Future opportunities exist to further constrain the generated water balance variables and their associated errors within process-based models.

## 1 Introduction

Under a changing climate and anthropogenic activities (Wada et al., 2010), it becomes urgent to accurately estimate the water balance components. Although distant from the Earth's surface, remote sensing (RS) satellites can uniquely provide information that translates into such estimates. For example, microwave and infrared techniques are used to estimate precipitation (Sun et al., 2018), evaporation can be computed using optical and thermal imagery (Zhang et al., 2016a), while changes in total water storage are obtained from the Gravity Recovery and Climate Experiment (GRACE) satellites (Wahr et al., 2004; Rodell et al., 2009). These advancements have emerged as powerful decision-support tools, pushing hydrology in



new directions by providing valuable information on the spatial distribution of water availability, use, and consumption (Hessels et al., 2022; Sheffield et al., 2018; Karimi et al., 2013). Such information is often challenging to obtain through traditional in situ data collection. With the rapid increase in RS data availability, it is now theoretically possible to close the

water balance. However, two challenges prevent this from being achieved, including the inconsistencies among various RS products for the same variable and the imbalances resulting from integrating the unique combinations of different water balance variables. These incoherences arise from the inherent data errors. Discrepancies in RS data, for example, come from different sources, such as errors in the retrieval algorithms (Maggioni et al., 2022), input data and parameters (Crosetto et al., 2001), or scale mismatch (Foken, 2008). While in situ data are known for their greatest reliability, they also come with their own errors.

Uncertainties in river discharge, for example, could be related to flow conditions (Mcmillan et al., 2012), velocity sensors (e.g., calibration) (Horner et al., 2018), or estimation methods such as the rating curve (Kuczera, 1996). Benefiting from the complementary strengths of in situ and RS data and embracing their errors requires a three-pronged methodology. This methodology should maximize the prior information content of the data available from multiple RS sources for the same variable, quantify both systematic and random errors in each water balance term, and reduce these errors by exploiting all

available information to generate calibrated estimates of spatially distributed water balance data. The following paragraphs review existing approaches in the literature used to handle these aspects.

Studies addressing the uncertainties in RS water balance data often rely on in situ data (treated as "ground-truth") to evaluate and converge on the different data products. However, in situ data (e.g., evaporation) is sparsely measured, and even when these data exist, accessibility may be challenging. Under such conditions, alternative indirect approaches such as triple

collocation (Tian and Peters-Lidard, 2010; Long et al., 2014; Massari et al., 2017; Yin and Park, 2021), uncertainty propagation (Hong et al., 2006; Cawse-Nicholson et al., 2020), and sensitivity analyses (Sobol, 2001) can be utilized. Other methods involve using the deviation from the ensemble mean, and the variability between the data sets as a proxy for uncertainty (Tian and Peters-Lidard, 2010; Sahoo et al., 2011; Munier et al., 2014; Zhang et al., 2018). Each of the previously summarized error estimation techniques has its inherent merits and shortcomings. In an attempt to capitalize on these strengths, Mourad et al.

(2024) recently integrated limited in situ data and multiple error metrics, along with expert judgment, to quantify and partially reduce water balance data errors. However, the resulting water balance estimates require further conditioning on in situ data using additional constraining steps, which is a subject of the methodology section of the current study.

Several other studies went beyond quantifying uncertainties in RS data and sought to fully reduce them using the water balance as a constraint (Aires, 2014; Hobeichi et al., 2020; Luo et al., 2023; Munier et al., 2014; Pan et al., 2012; Pan and Wood, 2006;

Rodell et al., 2015; Sahoo et al., 2011; Zhang et al., 2018; Zhang et al., 2016b). The core idea of the existing closure methods is to either select a preferred single data set of each water balance variable based on prior knowledge about their quality, or merge multiple data for each water balance variable using fixed pre-quantified error estimates. This step is followed by a water balance correction method such as variational data assimilation (L'ecuyer and Stephens, 2002), constrained Kalman filter (Pan and Wood, 2006), or proportional redistribution (Abolafia-Rosenzweig et al., 2021). In this process, the values of the

(un)merged water balance estimates are adjusted proportionally to their relative uncertainties or magnitudes, depending on the



correction method used. Contrary to the previously summarized water balance closure approaches, which fixed data errors a priori, Schoups and Nasseri (2021) proposed improving data error estimation by treating them along with the water balance variables as unknown random variables. Instead of enforcing the closure sequentially, the authors demonstrate that the water balance variables can be calibrated simultaneously in a single probabilistic data fusion methodology. The underlying premise

is that all variables are interconnected through the water balance. For instance, to calibrate precipitation data, the proposed approach uses statistical inference techniques to automatically fuse and adjust two information sources available on this variable, one coming from multiple estimates of precipitation RS data (original prior data), and the other sourced from other water balance data (predicted from the water balance). These techniques include an iterative smoother that involves multiple forward-backward passes over the timeseries. In the forward pass, the estimated precipitation carries information from previous

and current months, while in the backward pass it holds information from the future months. A similar process happens when estimating the other water balance variables, yielding refined posterior estimates that combine information from all other months.

Further to the above literature, previous attempts have been made to extend the basin-wide water balance closure constraints to a finer scale of up to 0.25° pixel resolution (Barkhordari et al., 2025; Heberger et al., 2023; Pellet et al., 2019; Munier et al.,

2014). Building on these previous efforts, our emphasis here is on generating detailed calibrated precipitation (0.05° resolution) and evaporation (250 meters resolution) estimates that align with the basin-scale closure constraints. To that end, we intend to contribute to the literature by extending the existing basin-scale water balance data fusion method first introduced by Schoups and Nasseri (2021). The extended methodology presented here retains the original model's key advantages, namely: the ability to simultaneously calibrate the basin-wide water balance variables by exploiting data from the entire timeseries. An integral

part of this new version is using the basin-wide posteriors to generate calibrated ensembles of spatially distributed precipitation and evaporation. This is achieved in a stepwise procedure, with the first step involving the formulation of ensemble-driven grid-scale error models, contrary to the models in Schoups and Nasseri (2021), which were formulated at the basin-scale and relied on an ensemble of two data sets. The second step is a constraint step carried out by the water balance data fusion at the basin scale, yielding basin-scale posteriors for all water balance variables, and a Kalman smoothing algorithm, yielding grid-

scale posteriors for precipitation and evaporation. This approach ensures consistency across the two scales: when back-calculating the basin-scale estimates from the grid-scale posteriors via spatial averaging, we recover the inferred basin-scale posteriors from the water balance data fusion. Lastly, the new version of the methodology incorporates the following additional features: (i) accounting for surface water imports from irrigation canals, which might significantly affect the overall water budget in irrigated basins, and (ii) quantifying posterior cross-correlations between water balance variables, which is important

for generating joint posterior samples. We apply the extended approach to an irrigated monsoon-influenced Hindon Basin, which suffers from unsustainable water use.

The remainder of this paper is structured as follows: section 2 introduces the case study and describes the data sets used as input for the probabilistic water balance model presented in Sect. 3. Section 4 details how all unknowns in the water balance model are solved at the basin scale, yielding a closed water balance, and how these water balance constraints are transferred



to the grid scale. The basin and grid-scale results are then presented in Sect. 5. The sensitivity of the water balance data fusion approach to the modeling decisions is then discussed in Sect. 6. Throughout the paper, we use the terms "priors" and "posteriors" to refer to the variables (i.e., water balance terms and error parameters) before and after calibration, respectively (i.e., before and after introducing the monthly water balance constraints).

## 2 Case Study: Hindon Basin in Northern India

The study focuses on the Hindon Basin in the northwestern Uttar Pradesh State of India (see Fig. 1). The Hindon basin covers a total area of about 4780 km$^2$ and drains to the Galeta outlet, where a river discharge station is located. The rainfed Hindon River originates from the Shivalik Hills at the foothills of the Himalayas. It flows towards the south on the flat and alluvial surfaces of the basin. The river eventually drains into the Yamuna River (near Noida, downstream of Delhi), which, in turn, joins the Ganga River before it reaches the Bay of Bengal. The basin features a large-scale supply-based irrigation system with

two major irrigation canals: the Eastern Yamuna Canal (EYC) and the Upper Ganga Canal (UGC). These canals are fed by two external reservoirs located outside the basin, namely the Bhimgoda Barrage constructed on the Ganga River and Hathni Kund Headwork on the Yamuna River. The share of the external canal water imports in the water balance is significant for this heavily irrigated basin, representing about 30-40% of the annual precipitation (Fig. A1). In addition to supplying surface irrigation water to the study area, the canals contribute to groundwater recharge through seepage, forming several groundwater

mounds near their vicinity. Hence, the canals approximately coincide with groundwater divides, limiting lateral groundwater flow across the eastern and western boundaries of the Hindon basin (Umar et al., 2008).

The canal system is operated in two irrigation seasons: the Kharif and Rabi seasons. Kharif season extends from March to September, in which the main irrigated crops are sugarcane and rice, and the Rabi season spans from October to March of the following year, where the principal crops are wheat and mustard. The irrigated croplands represent more than 85% of the basin,

while only about 10% of the basin's area is covered by non-croplands (Fig. A2). Irrigation water is diverted from the canals to supply the basin through off-takes (Fig. 1).



**Figure 1: Location of the Hindon basin in the Uttar Pradesh state of India (inset map); with a detailed view of the basin featuring its boundaries, topographic profile, location of the outlet, and the irrigation scheme with reservoirs and canal system (main map). Topographic basemap sources: Esri, USGS, FAO, NPS, GIS user community, and others.**

## 3 Probabilistic water balance model

Our goal is to estimate each term in the monthly water balance, written as:



$$S_t = S_{t-1} + \bar{P}_t - \bar{E}_t - Q_t + C_t \qquad (1)$$

Equation (1) combines the different water balance terms together, where $S_{t-1}$ and $S_t$ are the total water storage in the basin at the start and end of month $t$ (including surface storage, soil moisture, and groundwater), $\bar{P}_t$ and $\bar{E}_t$ are the basin-average

precipitation and evaporation (including transpiration), $Q_t$ is the river discharge at the basin outlet for month $t$, and $C_t$ is the total canal water import into the basin (sum of all intakes, see Fig. 1) for month $t$. The net lateral groundwater flows into or out of this basin are assumed to be negligible due to their small magnitude relative to the other water balance variables (Alam and Umar, 2013). All terms in Eq. (1) are expressed in units of mm of equivalent water depth. The following paragraphs describe the water balance data for the case study.

Each water balance component in Eq. (1) is independently derived, gathered, or measured either from satellite data, ground-based measurements, or both (Table 1). In the prior selection of the most reliable precipitation and evaporation ensembles (see: Mourad et al. (2024)), we followed other water balance closure studies that emphasize the use of earth observations over model outputs to minimize the effect of their related assumptions, except for the Multi-source Weighted-Ensemble Precipitation (MSWEP v2.8) product (Beck et al., 2019). MSWEP is an "optimal merging" of gauge observations, satellite observations,

and reanalysis model output. Along with MSWEP, we use two other monthly gridded precipitation ensemble members: the Tropical Rainfall Measuring Mission (TRMM) (Huffman et al., 2007) and the NASA/JAXA Global Precipitation Measurement (GPM-IMERG) (Huffman et al., 2019). By combining precipitation radar, passive microwave, and infrared satellites with ground-based observations from the Global Precipitation Climatology Centre (GPCC), TRMM and GPM-IMERG provide a comprehensive precipitation estimate over a given area. Compared to single-band radar on TRMM, GPM dual-frequency

precipitation radar provides a broader range of measurable precipitation rates and a better estimate of precipitation particle size (Hou et al., 2014). Spatially interpolated rain-gauge for the basin from Indian Meteorological Department (IMD) data set (Pai et al., 2014) is included in this analysis for comparison but not used in the model.

For gridded evaporation ensembles, we incorporate five members with diverse methodological approaches for estimating evaporation from remote sensing. One ET data set based on two-parallel Penman-Monteith (PM) models for both canopy and

soil: the pyWaPOR-ET v2.6 (Bastiaanssen et al., 2012), two ET data sets that estimate ET from a single-source surface energy balance (SEB) model representing vegetation and soil in a combined energy balance, namely the Landsat Collection 2 Provisional Actual Evapotranspiration Science Product (Landsat-based SSEBop ET) (Senay, 2018; Senay et al., 2023) and the Surface Energy Balance Algorithm for Land (eeSEBAL product) (Bastiaanssen et al., 1993), in addition to a two-source SEB ALEXI-ET data set from the Atmosphere-Land Exchange Inverse model (Anderson et al., 2007; Anderson et al., 2015). We

also incorporate the CMRSET ET product based on the CSIRO MODIS Reflectance-based Scaling ET model (Guerschman et al., 2009) that uses vegetation indices and meteorological data for scaling the ET. All data sets are resampled to the same spatial resolution using bilinear interpolation. More details on the theoretical background and processing steps for the P and E data sets can be found in the companion paper (Mourad et al., 2024).

The total water storage variable data were obtained from the recent release of the monthly Jet Propulsion Laboratory (JPL-

RL06) mascon solution. Each monthly total water storage estimate represents the surface mass anomaly relative to the baseline



average over 2004-2009. This version relies on prior geophysical information to constrain the solution, eliminating the need for empirical de-striping filtering commonly used for post-processing traditional spherical harmonic gravity solutions. However, intrinsic to this product are biases, that is, leakage errors, and spatial smoothing that damps the "true" signal, especially in small-sized basins. A Coastline Resolution Improvement (CRI) filter has been applied in this version to reduce

signal leakage errors across coastlines (Wiese et al., 2016). This data is also accompanied by scaling factors for optionally restoring the damped signal. However, these were not applied to the data used herein; instead, bias along with noise variance are modeled using all water balance data incorporated in this study (see Sect. 3.3).

The in situ data used in this analysis are obtained from governmental agencies, including the Central Water Commission (CWC) of India for streamflow data and the irrigation department of Uttar Pradesh for external canal water imports data. While

river discharge data has no gaps, canal delivery data are constrained by their spatial and temporal coverage. The latter comes in two forms: irrigation schedules and actual flow measurements. We apply an extrapolation approach that combines both data sources to generate complete monthly estimates for all intakes (for more details on the adopted gap-filling technique, refer to Appendix A2).

**Table 1: Monthly Water Balance Data. The data type column distinguishes between Remote Sensing (RS) and ground-based**
**Measurements (GBM).**

| Variable | Data Type | Data Source | Original (Resampled) resolution | Study period |
|---|---|---|---|---|
| Precipitation $(P_{obs})$ | RS and GBM | TRMM TMBA (TRMM3B43 v7) | 0.25° (0.05°) | 2003-2019 |
| | RS and GBM | GPM (IMERG) 3IMERGDF (v06) | 0.1° (0.05°) | 2003-2022 |
| | Merged | MSWEP v2.8 | 0.1° (0.05°) | 2003-2022 |
| Evaporation $(E_{obs})$ | RS | pyWaPOR v2.6 | 250m (250m) | 2003-2022 |
| | RS | Landsat C2 Provisional ET (Landsat-SSEBop) | 30m (250m) | 2003-2022 |
| | RS | eeSEBAL | 30m (250m) | 2003-2022 |
| | RS | CMRSET | 5km (250m) | 2003-2012 |
| | RS | ALEXI | 5km (250m) | 2003-2015 |
| Storage $(S_{obs})$ | RS | GRACE JPL mascon (RL06 v1.0) | 3° (basin-wide) | 2003-2022 |
| River discharge $(Q_{obs})$ | GBM | Stream gauge (Central Water Commission, 2024) | point | 2003-2022 |
| Canal Water Imports $(C_{obs})$ | GBM | Canal gauges and irrigation schedules (Irrigation & Water Resources Department, 2024) | distributary-based | 2003-2022 |





Despite the variety of the above-explored data sources, two barriers hinder their potential use in hydrological applications. One occurs when integrating the unique data combinations describing each water balance variable into the basin-scale water balance. In any particular month, we end up with significant errors, i.e., either too much or too little water, resulting in a non-zero net balance (Fig. B1 in Appendix B). This non-closure or imbalance problem highlights the incoherences between the

water balance data sets and the inconsistencies in the individual data sets, caused by various data errors. Another well-known challenge is the inconsistencies between the various data products for the same variable. For example, for the studied basin, grid-scale inter-product uncertainty is moderate for $P$, while $E$ exhibits significant spatial variability across the whole basin (Mourad et al., 2024). These challenges motivate the distinction between the "calibrated" value of each water balance variable that satisfies the water balance closure at the basin scale and represents the error-corrected estimate at the grid level,

and the "observed" values, which deviate from these values. In the probabilistic form of the water balance, each term is assigned a probabilistic data error model to quantify systematic and random differences between observed and modelled water balance variables. Parameters in these data error models are treated as unknown random variables with predefined prior distributions. In this regard, the overall probabilistic water balance takes the form of a Bayesian hierarchical model with two levels of uncertainty: one related to the error parameters and another linked to the water balance variables (Schoups

and Nasseri (2021)). The following subsections describe the parametric probabilistic relations adopted for each water balance variable ($P$, $E$, $Q$, $C$, and $S$).

**3.1 Precipitation and Evaporation error models**

Contrary to Schoups and Nasseri (2021), we define prior error models for $P$ and $E$ at the grid scale instead of at the basin scale. As described below, spatial averaging is used to derive corresponding priors at the basin scale. The latter are then used

with water balance data fusion for estimating basin-scale posteriors (Sect. 4.1). The resulting basin-scale posteriors are finally transferred back to the grid scale (Sect. 4.2).

The first step in setting up a prior error model for the gridded $P$ and $E$ variables is to probabilistically characterize them with prior distributions. For these two independent spatial processes, we can write their grid-scale joint prior distribution for each month $t$ as:

$$\mathcal{N}\left(\begin{bmatrix} P_t \\ E_t \end{bmatrix} \middle| \begin{bmatrix} m_{P_t} \\ m_{E_t} \end{bmatrix}, \begin{bmatrix} V_{P_t} & 0 \\ 0 & V_{E_t} \end{bmatrix}\right) \tag{2}$$

where $P_t$ and $E_t$ are vectors containing unknown precipitation and evaporation values for all grid cells in the corresponding spatial field for each month $t$. The Gaussian distribution (Eq. (2)) is specified with a mean vector and a block diagonal covariance matrix with zero cross-covariance to reflect that these two processes are assumed to be independent a priori. The mean vector consists of the prior means of $P$ ($m_{P_t}$) and $E$ ($m_{E_t}$), for all spatial locations s of each month $t$, while the autocovariance of each variable $P$ ($V_{P_t}$) and $E$ ($V_{E_t}$) is the block-diagonal element of the covariance matrix. The

decomposed form of the joint distribution for each individual variable is described in the following subsections.



### 3.1.1 Precipitation

The independence structure between $P_t$ and $E_t$ shown in Eq. (2) allows us to represent the errors in each variable separately. For each month $t$ and at each grid cell location s, we typically have multiple measurements of precipitation, with unknown bias and random errors. To characterize these errors, the grid-scale precipitation bias and random error models for each

element in the mean vector $m_{P_t}$ and diagonal entries of the autocovariance matrix $V_{P_t}$ are defined using Eqs. (3-4), respectively:

$$m_{P_{s,t}} = P_{s,t}^{obs,min} + w_p\left(P_{s,t}^{obs,max} - P_{s,t}^{obs,min}\right) \tag{3}$$

$$s_{P_{s,t}} = r_P \frac{1}{4}\left(P_{s,t}^{obs,max} - P_{s,t}^{obs,min}\right) \tag{4}$$

Equation (3) models the systematic bias in precipitation at each location s for each month $t$ by describing the precipitation prior mean $m_{P_{s,t}}$ as a function of two parts. The first part represents the baseline precipitation estimated as the minimum across all grid cells for the location s ($P_{s,t}^{obs,min}$), while the second consists of a weighted deviation from this base value. Specifically, the

latter denotes the full range of the grid-wise observed precipitation ($P_{s,t}^{obs,max} - P_{s,t}^{obs,min}$) with a weight parameter $w_p$. The bias parameter $w_P$ takes on an unknown value between 0 and 1, with a logit-normal prior distribution of parameter $\mu = 0$ and scale parameter $\sigma = 1.4$ to reflect prior uncertainty about the bias. In other words, $w_P$ controls the relative position of the $m_{P_{s,t}}$ within the observed precipitation space (range). As it approaches 0, more weight is given to $P_{s,t}^{obs,min}$, whereas approaching a value of 1 gives more weight to $P_{s,t}^{obs,max}$.

Random errors in the precipitation are modeled using Eq. (4). This model expresses the prior standard deviation $s_{P_{s,t}}$ at grid cell location s for each month $t$ as a function of the maximum potential random error, defined as a quarter of the range: $\frac{1}{4}\left(P_{s,t}^{obs,max} - P_{s,t}^{obs,min}\right)$. A noise parameter ($r_P$) is used to scale this conservative quantity and is given a quasi-uniform prior distribution between 0 and 1.

To account for the effect of spatial correlation of the random error component (Eq. (4)), we write the precipitation prior

covariance matrix $V_{P_t}$ in terms of the grid-scale standard deviations and a grid-scale auto-correlation matrix:

$$V_{P_t} = S_P R_P S_P \tag{5}$$

where $S_P$ is a diagonal matrix containing the grid-scale $s_{P_{s,t}}$ values for all locations of the spatial field (Eq. (4)), and $R_P$ is the correlation matrix that captures the spatial dependence structure. $R_P$ is jointly estimated from all precipitation data using an isotropic parametric correlation function with the following form (Handcock and Stein, 1993):

$$C_{\mathcal{M}}(d|l_s, v) = \frac{1}{2^{v-1}\Gamma(v)}\left(\frac{d}{l_s}\right)^v K_v\left(\frac{d}{l_s}\right) \tag{6}$$

where $C_{\mathcal{M}}$ is the Matérn correlation function for variables separated by distance $d$. This correlation model is flexible and

widely used, with two functions: gamma function $\Gamma(.)$ and the modified Bessel function $K_v(.)$ (Abramowitz and Stegun,



1968). $C_{\mathcal{M}}$ also consists of two unknown nonnegative parameters, namely the spatial correlation length scale $l_s$ and a spatial smoothness parameter $\nu$. A value of $\nu$ approaching 0 indicates a rough spatial process, while the process is smoother when $\nu$ approaches infinity. Since the smoothness parameter is usually small in many applications (Chen et al., 2022), while it increases as the aggregation time increases (Sun et al., 2015), we choose a balanced value between a rough and smooth random field,

i.e., $\nu$ , fixed at 1.5. On the other hand, the correlation length scale ($l_s$ ) defines an average length scale on which grid cells are correlated with each other. In principle, $l_s$ ranges from 0 (the case of uncorrelated grid cells) and extends to a scale larger than the spatial domain length (the case of maximally correlated pixels). With no prior information on the $l_s$ parameter, we fix it at 50 km (~1/2 the basin's length from North to South). The sensitivity of the results to the fixed $l_s$ will be evaluated in section 6.2.

Since water balance data fusion (Schoups and Nasseri, 2021) uses basin-scale error models, we derive these from the above-described grid-scale models by spatial averaging. Specifically, the basin-scale prior mean $m_{\overline{P_t}}$ and variance $v_{\overline{P_t}}$ in month $t$ are given by:

$$m_{\bar{P}_t} = \phi_P' m_{P_t} = \overline{m}_{p_t} \tag{7}$$

$$v_{\bar{P}_t} = \phi_P' V_P \phi = \bar{V}_{P_t} \tag{8}$$

where $\overline{m}_{p_t}$ is the average of all elements in $m_{P_t}$, $\bar{V}_{P_t}$ is the average of all elements in the matrix $V_{P_t}$, and $\phi_P$ is the spatial averaging operator used to derive basin-scale moments from grid-scale moments (i.e., $s \times 1$ vector with each element equal to

$1/s$ , where $s$ is the number of spatial locations in the precipitation spatial field).

Substituting the grid-scale models specified above, Eqs. (3-4), into the spatial averaging, Eqs. (7-8), yields the following basin-scale error model for precipitation:

$$m_{\bar{P}_t} = \overline{P_t^{obs,min}} + w_p \overline{\left(P_t^{obs,max} - P_t^{obs,min}\right)} \tag{9}$$

$$s_{\bar{P}_t} = r_P \sqrt{\overline{D_P R_P D_P}} \tag{10}$$

$$\bar{P}_t \sim \mathcal{N}\left(m_{\bar{P}_t}, s_{\bar{P}_t}^2\right) \tag{11}$$

$$\bar{P}_t \geq 0 \tag{12}$$

where $D_P$ is a diagonal matrix containing the $\frac{1}{4}\left(P_{s,t}^{obs,max} - P_{s,t}^{obs,min}\right)$ values and overbars denote spatial averaging (e.g., $\overline{P_t^{obs,min}}$ with a dropped s subindex, is the spatial average across all grid cell values within the minimum precipitation field).

All overbar (basin-averaged) quantities are precomputed from the precipitation data sets, and the constant but unknown parameters $w_p$ and $r_P$ are estimated as part of the water balance data fusion (see Sect. 4.1).





Finally, the last two equations in the precipitation error model treat the basin-scale calibrated precipitation $\bar{P}_t$ for each month $t$ as a random draw from a truncated normal distribution. The truncation at zero ensures physical consistency (nonnegative precipitation).

### 255  3.1.2 Evaporation

As with precipitation, an evaporation error model with the following range-based form is adopted:

$$m_{E_{s,t}} = f_E \left[ E_{s,t}^{obs,min} + w_E \left( E_{s,t}^{obs,max} - E_{s,t}^{obs,min} \right) \right] \tag{13}$$

$$s_{E_{s,t}} = r_E \frac{1}{4} \left( E_{s,t}^{obs,max} - E_{s,t}^{obs,min} \right) \tag{14}$$

The bias in evaporation is modeled with two spatial and time-invariant calibration parameters, namely: $w_E$ and $f_E$. The parameter $w_E$ is the weight that interpolates between the monthly evaporation extrema $E_{s,t}^{obs,min}$ and $E_{s,t}^{obs,max}$ at each grid cell location s and month $t$. An additional scaling factor ($f_E$) is incorporated to account for potential bias outside the observed

range. On the other hand, Eq. (14) quantifies evaporation prior uncertainty ($\frac{1}{4} \left( E_{s,t}^{obs,max} - E_{s,t}^{obs,min} \right)$) scaled via the spatially and temporally constant noise parameter ($r_E$). All parameters are treated as random variables with prior distributions. The $w_E$ and $r_E$ parameters are specified with a vague prior distribution bounded between 0 and 1, specifically, flat logit-normal with location parameter $\mu = 0$ and scale parameter $\sigma = 1.4$. Whereas, $f_E$ is given a lognormal prior with mode at 1 (no bias) and a coefficient of variation CV of 50%. Notably, all unknown parameters are solved as part of the water balance data fusion (Sect.

265  4).

The basin-scale error models are derived from the grid-based models defined above, using the same spatial averaging process applied to precipitation (Eqs. (7-8)). The averaging formulas are then obtained as follows:

$$m_{\bar{E}_t} = f_E \left[ \overline{E_t^{obs,min}} + w_E \overline{\left( E_t^{obs,max} - E_t^{obs,min} \right)} \right] \tag{15}$$

$$s_{\bar{E}_t} = r_E \sqrt{\overline{D_E R_E D_E}} \tag{16}$$

$$\bar{E}_t \sim \mathcal{N} \left( m_{\bar{E}_t}, s_{\bar{E}_t} \right) \tag{17}$$

$$\bar{E}_t \geq 0 \tag{18}$$

where $D_E$ is a diagonal matrix whose diagonal entries are $\frac{1}{4} \left( E_{s,t}^{obs,max} - E_{s,t}^{obs,min} \right)$ grid-based values within the spatial domain, $\overline{E_t^{obs,min}}$, $\overline{\left( E_t^{obs,max} - E_t^{obs,min} \right)}$ and $\sqrt{\overline{D_E R_E D_E}}$ are the prior quantities computed from the evaporation data sets through

spatial averaging. The $R_E$ term in the last quantity stands for the correlation matrix, which captures the spatial dependencies between the evaporation grid cells. For the large-sized evaporation data sets considered here, we parameterize the evaporation correlation matrix $R_E$, here, using a Matérn kernel implemented within a stochastic variational Gaussian Process (Hensman et al., 2015) with fixed parameter $l_s$ at 50 km and $\nu$ at 1.5.





Similar to precipitation, the basin-scale calibrated evaporation $\bar{E}_t$ for month $t$ is treated as a random draw from a truncated
normal distribution Eqs. (17-18). The truncation at zero ensures physical consistency (nonnegative evaporation).

### 3.2 River discharge and canal water import error models

We use the same proportional error model to relate the river discharge ($Q_{obs}$) and canal water imports ($C_{obs}$) observations to
their modeled values. The model takes the following general form:

$$m_{x,t} \sim \mathcal{N}\left(x_{obs,t}, v_{obs,t}\right) \tag{19}$$

$$s_{x,t} = a_x x_{obs,t} + b_x \tag{20}$$

$$x_t \sim \mathcal{N}\left(m_{x,t}, s_{x,t}^2\right) \tag{21}$$

$$x_t \geq 0 \tag{22}$$

where $x_{obs}$ represents the variable of interest ($Q_{obs}$ for discharge and $C_{obs}$ for canal water imports). Given the complete $Q_{obs}$
records and gap-filled $C_{obs}$ data, for each month, we set $v_{obs,t} = 0$, such that $m_{x,t}$ becomes equal to the observed (unbiased)
data for that specific month. The standard deviation $s_{x,t}$ represents the random errors in the $Q_{obs}$ and $C_{obs}$. These errors are
modeled as a linear function defined by two time-invariant parameters: $a_x$(slope) and $b_x$(intercept). This suggests that as the
magnitude of the variable of interest increases, its potential random error also increases. We assume tight prior constraints on
$a_x$ and $b_x$ to reflect the strong prior belief on these measurements. Namely, $a_x$ is assigned a lognormal prior with a mode at
0.1 for $Q_{obs}$ and 0.25 for $C_{obs}$ (i.e., a relative error of 10% and 25%, respectively) and a small CV of 1%, while $b_x$ is assigned
a lognormal prior with mode at 0.001 and also a CV of 1%. Schoups and Nasseri (2021) showed that these priors provide
reasonable approximations of $a_x$ and $b_x$ for basins where the measured variable (here: $Q_{obs}$ and $C_{obs}$) is small in magnitude
relative to the other water balance variables.

### 3.3 Water Storage Error Model

A key feature of the water storage error model is that it relates the storage observations from the GRACE satellite ($S_{obs,t}$) to
the corresponding calibrated storage ($S_t$), instead of the other way around (as in the earlier defined error models). Due to the
coarse resolution of GRACE data, the monthly basin-scale total water storage derived from these observations can be polluted
by the water storage dynamics occurring outside the basin, that is, "leakage". To account for the temporal and spatial mismatch
between GRACE basin-scale water storage and the calibrated storage caused by leakage errors, we employ the following noisy
sine wave error model (Schoups and Nasseri, 2021):

$$m_{S,t} = S_t + A \sin\left(\omega\left(\frac{t}{12} - \delta\right)\right) \tag{23}$$



$$s_{S,t} = \sigma_S \tag{24}$$

$$S_{obs,t} \sim \mathcal{N}(m_{S,t}, s_{S,t}^2) \tag{25}$$

The first equation models the systematic errors associated with GRACE observations. It captures cyclic patterns and seasonality in the data within the basin via time-invariant error parameters. These parameters describe the magnitude and the timing of the seasonal error: the amplitude $A$ (mm) and phase $\delta$ (years), respectively.

The second equation models the magnitude of random errors as an unknown time-invariant parameter, which reflects errors that can arise from (a) any limitations caused by the sine wave models in capturing the calibrated water storage dynamics within the basin and (b) noise in GRACE solutions. In the above equations, $\omega$ is fixed at $2\pi$ radians per year, yielding a one-year sine wave and thus capturing the annual seasonal cycle. Other parameters are assigned vague prior distributions where $A$ follows a lognormal prior with mode at 30 mm and a CV of 200%, $\sigma_S$ a lognormal prior with mode equal to 10 mm and a CV of 200%, and $\delta$ follows a flat logit-normal prior between 0 and 1 year with a location parameter $\mu = 0$ and scale parameter $\sigma = 1.4$.

While the above prior error model relies on a single storage data set, we assess a posteriori (Sect. 6.1) the sensitivity of the $P$ and $E$ posterior distributions to the use of different storage inputs, including Mascon (JPL) and Spherical Harmonic (CSR) GRACE solutions with considerably varying magnitudes in long-term trends.

## 4 Inference methods

The following subsections detail the two-step statistical inference framework used to solve the basin-scale probabilistic water balance model (Sect. 4.1) and then propagate the resulting basin-wide constraints to the grid level (Sect. 4.2).

### 4.1 Basin-scale

Following the description of the prior probabilistic relations (Sect. 3), the posteriors of water balance variables and error parameters are computed using a basin-scale inference technique. This technique consists of a double-loop method that combines two algorithms: a single-chain differential evolution variant of Markov Chain Monte Carlo (DE-MCMC) for computing the unknown parameters and Expectation Propagation (EP) for solving the unknown water balance variables. A brief overview of these algorithms is described in the Appendix B, while a detailed explanation can be found in the original paper (Schoups and Nasseri, 2021). The computed posteriors provide hydrologically consistent water balance variables that jointly close the water balance. In the water balance data fusion process, the independent prior data (Sect. 3) are updated with the monthly water balance constraints. These constraints link all independent variables through the water balance, thereby inducing cross-correlation between them. For example, the term S in the water balance constraint (dropping the time index t for simplicity: $S = S_0 + \bar{P} - \bar{E} - Q + C$) linearly links all other terms. To maintain the water balance closure, the terms with opposite signs must covary together, and vice versa. Here, an additional post-processing step is applied to represent this



interdependence structure between the water balance variables, i.e., compute their covariances. For a vector of posterior variables $x$ (including the initial storage state $S_0$ or $S_{t-1}, \bar{P}, \bar{E}, Q$, and $C$), a joint distribution can be written for each month $t$ as: $p(x_t) \sim \mathcal{N}(m_{x,t}^*, V_t^*)$ with a mean vector $(m_{x,t}^*)$ holding the marginal posterior means of each variable (Eq. (B2)) and a $5 \times 5$ covariance matrix $V_t^*$ that encodes the relation between these variables. The diagonal element of this matrix contains each variable's marginal posterior variance, while the unknown covariances between each variable pair are the off-diagonal entries. To quantify these covariances, a message-passing algorithm that combines the water balance constraint and the probabilistic constraints in the form of marginal Gaussian distributions is applied (See Text B2 in Appendix B).

## 4.2 Grid-scale

Having computed the joint basin-scale $\bar{P}$ and $\bar{E}$ posteriors (Sect. 4.1), the goal is to translate these posteriors back to the grid-scale. When spatially averaged, the generated grid-scale posteriors ($P$ and $E$) should recover the basin-scale posteriors. Such a relation between the two scales can be established for each month $t$ as follows (Eq. (26)):

$$\begin{bmatrix} \bar{P}_t \\ \bar{E}_t \end{bmatrix} = A \begin{bmatrix} P_t \\ E_t \end{bmatrix} \tag{26}$$

where $A = \begin{bmatrix} \phi_P' & 0 \\ 0 & \phi_E' \end{bmatrix}$ represents the spatial averaging operator mapping the grid-scale to the basin-scale P and E variables, $\phi_P$ is defined earlier in Eq. (8), while $\phi_E$ is a $s \times 1$ vector with each element equal to $1/s$, where $s$ here is the number of spatial locations in the evaporation spatial domain. $\phi_P'$ and $\phi_E'$ are their transpose.

For the above relation to hold (Eq. (26)), we implement a Kalman smoothing algorithm. A key aspect of this algorithm is that it ties together all sources of information on $P$ and $E$ variables, detailed in previous sections, including: the basin and grid-scale joint priors (Sect. 3), along with basin-scale joint posteriors (Sect. 4.1), to generate their corresponding spatially distributed joint posteriors. Distinctly, the grid-scale prior moments (Eq. (2)) are updated to reflect the difference between the basin-scale joint posterior and prior moments via the Kalman gain. In what follows, we start by describing the basin-scale joint prior and posterior distributions and then show how the grid-scale joint distributions are computed.

Before constraining the basin-scale spatially averaged $\bar{P}_t$ and $\bar{E}_t$ priors, their joint distribution for each month $t$ can be written with zero-cross covariances to reflect the independent structure between both variables:

$$\mathcal{N}\left(\begin{bmatrix} \bar{P}_t \\ \bar{E}_t \end{bmatrix} \middle| \begin{bmatrix} m_{\bar{P}_t} \\ m_{\bar{E}_t} \end{bmatrix}, \begin{bmatrix} v_{\bar{P}_t} & 0 \\ 0 & v_{\bar{E}_t} \end{bmatrix}\right) \tag{27}$$

where $m_{\bar{P}_t}$ and $m_{\bar{E}_t}$ are prior means, $v_{\bar{P}_t}$ and $v_{\bar{E}_t}$ are prior variances of $\bar{P}_t$ and $\bar{E}_t$, respectively.

Once the water balance constraint is imposed within the basin-scale data fusion: $\bar{P} - \bar{E} - Q + C + S_0 - S = 0$ (omitting t for simplicity), positive correlations between $\bar{P}$ and $\bar{E}$ posteriors emerge. This occurs because the water balance acts as a constraint on the priors (independent $\bar{P}$ and $\bar{E}$) linearly linking them in their posteriors (correlated $\bar{P}$ and $\bar{E}$). To maintain this linear constraint, that is, $\bar{P} - \bar{E}$ remains equal to $Q - C - S_0 + S$, an adjustment that increases $\bar{P}$ should be balanced by an



adjustment that increases $\bar{E}$ , and vice versa. This yields a basin-scale joint posterior distribution on the posterior variables $\bar{P}_t$ and $\bar{E}_t$, with induced correlations, that is, non-zero cross covariances:

$$\mathcal{N}\left(\begin{bmatrix}\bar{P}_t \\ \bar{E}_t\end{bmatrix} \Big| \begin{bmatrix}m_{\bar{P}_t}^* \\ m_{\bar{E}_t}^*\end{bmatrix}, \begin{bmatrix}v_{\bar{P}_t}^* & v_{\overline{PE}_t}^* \\ v_{\overline{PE}_t}^* & v_{\bar{E}_t}^*\end{bmatrix}\right) \tag{28}$$

where $m_{\bar{P}_t}^*$ and $m_{\bar{E}_t}^*$ are the marginal posterior means, $v_{\bar{P}_t}^*$ and $m_{\bar{E}_t}^*$ are the marginal posterior variances of $\bar{P}_t$ and $\bar{E}_t$, respectively, and $v_{\overline{PE}_t}^*$ is the posterior cross-covariance between $\bar{P}_t$ and $\bar{E}_t$.

Kalman smoothing can then be used to calculate the joint posterior moments Eqs. (29-30) at the grid-scale from the above basin-scale prior moments Eqs. (27-28) and grid-scale prior moments (Eq. (2)). These posterior moments are thus rendered as a weighted combination of grid and basin-scale moments:

$$\begin{bmatrix}m_{P_t}^* \\ m_{E_t}^*\end{bmatrix} = \begin{bmatrix}m_{P_t} \\ m_{E_t}\end{bmatrix} + K\left(\begin{bmatrix}m_{\bar{P}_t}^* \\ m_{\bar{E}_t}^*\end{bmatrix} - \begin{bmatrix}m_{\bar{P}_t} \\ m_{\bar{E}_t}\end{bmatrix}\right) \tag{29}$$

$$\begin{bmatrix}V_{P_t}^* & V_{PE_t}^* \\ V_{PE_t}^{*\prime} & V_{E_t}^*\end{bmatrix} = \begin{bmatrix}V_{P_t} & 0 \\ 0 & V_{E_t}\end{bmatrix} + K\left(\begin{bmatrix}v_{\bar{P}_t}^* & v_{\overline{PE}_t}^* \\ v_{\overline{PE}_t}^* & v_{\bar{E}_t}^*\end{bmatrix} - \begin{bmatrix}v_{\bar{P}_t} & 0 \\ 0 & v_{\bar{E}_t}\end{bmatrix}\right)K' \tag{30}$$

where $m_{P_t}^*$ and $m_{E_t}^*$ are the grid-scale means, while $V_{P_t}^*$ and $V_{E_t}^*$ are autocovariances of $P_t$ and $E_t$ posteriors, respectively.

$V_{PE_t}^*$ and $V_{PE_t}^{*\prime}$ stand for the posterior cross-covariance between both variables and its transpose, respectively.

A noteworthy feature of the Kalman smoothing gain $(K)$ in the above equations is that it acts as a weighting matrix propagating basin-scale information to each individual $P$ and $E$ grid cell. It is computed using the following equation composed of three components:

$$K = \begin{bmatrix}V_{P_t} & 0 \\ 0 & V_{E_t}\end{bmatrix} A' \begin{bmatrix}v_{\bar{P}_t} & 0 \\ 0 & v_{\bar{E}_t}\end{bmatrix}^{-1} \tag{31}$$

The first component depicts the block diagonal covariance matrix of the grid-scale $P_t$ and $E_t$ priors, the second is the transpose

of the spatial averaging operator $A$, whereas the inverse of the diagonal matrix consisting of the basin-wide variances of $\bar{P}_t$ and $\bar{E}_t$ priors is the third component.

## 5 Results

Here, we present the results of the two-step inference framework described in section 4, starting with the basin-scale posteriors of all water balance variables (Sect. 5.1) and followed by the grid-scale posteriors of $P$ and $E$ (Sect. 5.2). These results are

based on specific choices of GRACE data (i.e., JPL mascon data) and spatial correlation lengths. The sensitivity of the results to these choices will be addressed in Sect. 6.





## 5.1 Basin-scale posteriors

Basin-scale posteriors for $P$ (precipitation), $E$ (evaporation), and $C$ (canal water imports) are shown in Fig. 2 and discussed first. The data shows significant prior uncertainty with a wide range of precipitation and evaporation estimates. This is

especially evident in the monsoon (June to September) for precipitation and across the whole seasonal cycle for evaporation (Fig. 2). Along with the prior observations, Fig. 2 also depicts the basin-scale posteriors for each water balance term, while Fig. 3 shows the corresponding error parameter posteriors. The results for $Q$ are not included in Fig. 2 since the posterior for $Q$ closely follows the river discharge data, and this data is tied to a non-disclosure agreement with the CWC of India.

The results in Fig. 2 show that the precipitation posterior mean lies between the $\bar{P}_t^{obs,min}$ and $\bar{P}_t^{obs,max}$ data bounds. This

balanced position within the precipitation space is also translated into the inferred parameter $w_P$, with a posterior distribution around values of $\sim 0.5$ (Fig. 3). The noise parameter $r_P$ has a posterior distribution around 1, suggesting a minimal reduction in the prior uncertainty range for precipitation. An independent evaluation of these posterior estimates can be done by comparing them to the Indian Meteorological Department (IMD) data set (Pai et al., 2014), which is routinely used to evaluate precipitation products in India. Despite being treated as the "ground truth" in literature, the rain-gauge interpolated IMD

product is not error-free, and therefore, a perfect match between the IMD data and the generated posterior mean is not anticipated. Figure C1 in Appendix C compares the seasonal and annual timeseries of the IMD to the posterior mean. Unlike the precipitation posterior mean, which tends to sit between observed data bounds, the IMD data set closely follows the baseline precipitation estimate (TRMM). The larger posterior estimates for $P$ found here align with a recent study (Goteti and Famiglietti, 2024) that found systematic underestimation of precipitation by the IMD data product.

Moving to evaporation, the computed posteriors in Fig. 2 only slightly differ from the priors, with posterior values of noise parameter $r_E$ centered at $\sim 0.95$ (Fig. 3), implying that the modeled random error in this variable is as large as one-fourth of the absolute difference between $\bar{E}_t^{obs,min}$ and $\bar{E}_t^{obs,max}$. This posterior uncertainty, however, reduces with a tighter data range corresponding to prior ensembles generated from only three data sets: eeSEBAL, pyWaPOR, and Landsat-SSEBop (specifically, from 2015 onwards). As for the estimated evaporation, it takes more or less a balanced weight between $\bar{E}_t^{obs,min}$

and $\bar{E}_t^{obs,max}$, with an estimated $w_E$ posterior of $\sim 0.48$ and a scaling parameter $f_E$ around 1 (no additional bias). While no direct evaporation measurements are available, the posterior estimates can be evaluated by comparing them with reference evapotranspiration and expected seasonal dynamics based on known cropping and irrigation practices. For example, the evaporation is expected to be lower than $\overline{ET_0}$ but equal to or higher than the $\overline{ET_0}$ for only a short period, either in June-July or August-October, when the (dominant) sugarcane crop coefficient (Kc) exceeds 1. In Fig. 2, the inferred posterior evaporation

(green band) aligns with this expectation where it is consistently below $\overline{ET_0}$, but is also around $\overline{ET_0}$ for the previously mentioned period (e.g., in 2006, 2012, 2019, and 2022). Besides sugarcane, rice is the second most important crop, commonly transplanted at the end of April or mid-May (Joseph and Ghosh, 2023). For this shallow-rooted crop, more frequent irrigation is required during its early stages (the first 30 days), coinciding with the period when the canal water supplies form a peak (at the onset of the monsoon) (Fig. C3 in Appendix C). The evaporation timeseries of rice is, thus, expected to form a peak, most





likely in June, when its Kc value is higher than 1. Considering these two elements, the evaporation is expected to peak in June, as captured by the posterior mean (Fig. C2 in Appendix C).

The third row of Fig. 2 compares the prior and posterior estimates of the canal water imports ($C$). The prior estimates represented here are extrapolated from scarce data. Nevertheless, the extrapolation approach mostly relied on ground-based measurements whenever available. For periods with missing data, these were filled with the design discharge capacity

multiplied by the operation time. In that way, the gap-filled full timeseries of the canal water imports can be seen as upper-bound estimates. The applied extrapolation approach also allows for annual variations in the canal water supply, by using independent information from the Palmer Drought Severity Index (PDSI) index of Terra Climate data set (Abatzoglou et al., 2018) as a proxy to classify the years between wet and dry. For example, in dry years, such as 2008, and 2016 to 2018, the surface water supplies are smaller in magnitude compared to all other years. Note also the increase in the water supplies from

2009 onwards, which aligns with the construction of the Deoband Parallel Canal that connects the Upper Ganga Canal (UGC) to the Eastern Yamuna Canal (EYC) (Fig. 1). Using other approaches to generate the prior canal water estimates does not significantly affect the posteriors of the other terms, as explored in Sect. 6.3. The posterior results, on the other hand, reflect the wide prior uncertainty (relative error of 25%) we place around this extrapolated data. Generally, the posterior estimates follow both their priors and the seasonality of the precipitation term. Specifically, the supplies peak at the onset of the monsoon

(June) due to increased water availability from precipitation, while smaller peaks are formed during the dry season (Dec and April) when water stored in the reservoirs is diverted for irrigation. As canal supplies are mainly driven by precipitation, it is reasonable to conclude that the canal water imports are smaller in magnitude compared to precipitation during the rainy months (July-Sept), representing about 5 to 15% of the monthly precipitation. It follows that the fixed prior uncertainty on this term is a justifiable approximation that does not strongly affect the other water balance estimates.

Finally, we note the interplay between all variables ($P$, $E$, $C$, and $Q$). Despite relying on external water supplies, the basin has been experiencing a decreasing trend in the river discharge (Dwivedi and Yadav, 2025), coupled with increased posterior evaporation mean from 2010 to 2022 (Fig. C2). The interplay is also pronounced in dry years such as 2009 and 2016, showing a decline in the posterior mean precipitation, together with the canal water imports. The limited availability of rainfall and surface water for irrigation during droughts does not, however, depress the evaporation rates because crop water demand is

met by increased groundwater pumping. The canal water imports is an order of magnitude smaller than evaporation, which also implies dependence on groundwater abstractions. The effect of unsustainable groundwater pumping is particularly notable in the dynamics and trend of basin water storage, which we discuss next.





**Figure 2: Monthly water balance posteriors (90% credible intervals in green) for *P* (precipitation), *E* (actual evapotranspiration), and *C* (canal water imports), and their corresponding observations (dots). Reference evapotranspiration computed here using ERA5 meteorological input variables is shown as ($ET_0$). The overbars of the labels in *P* and *E* plots indicate that these values are obtained through spatial averaging of the gridded data sets for each month *t*. Each year's label indicates the start of the year (January).**





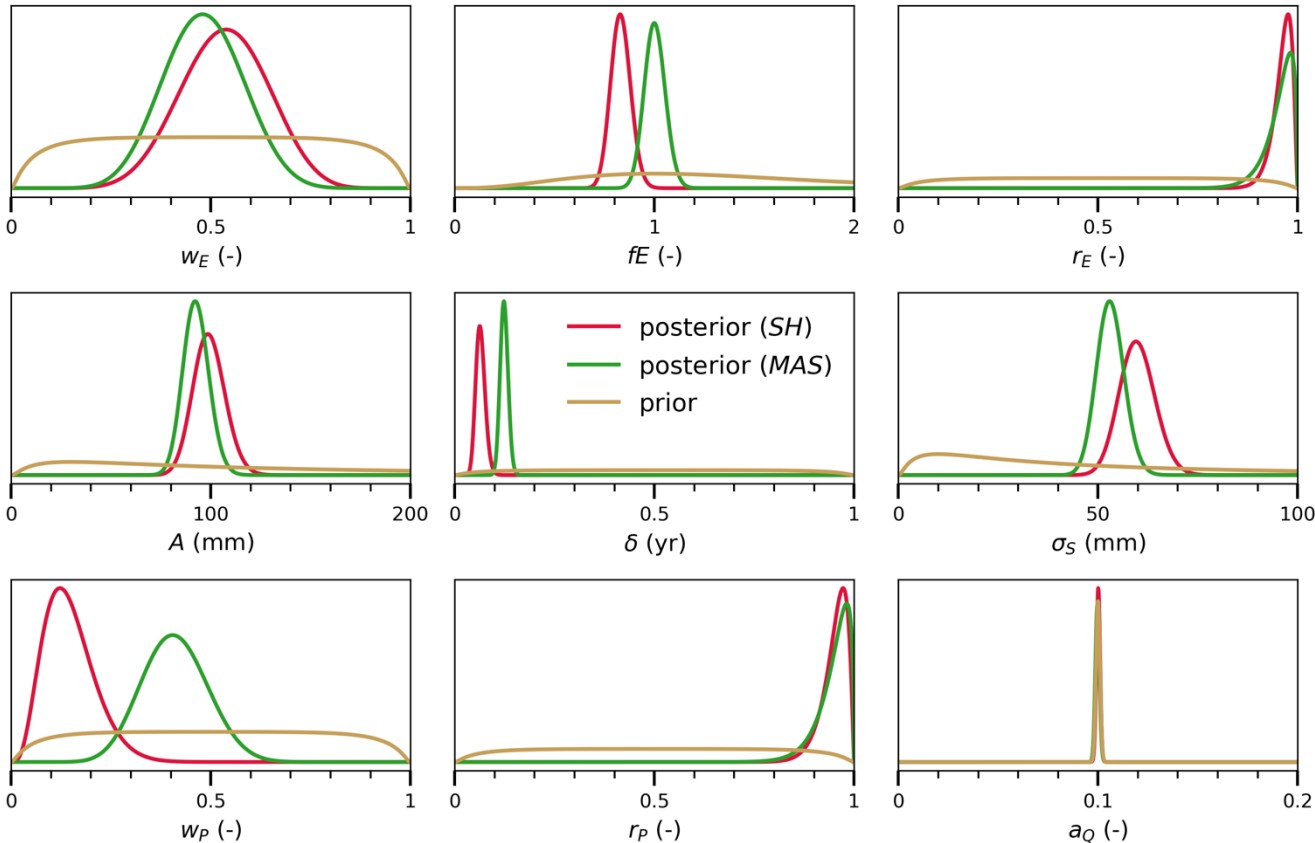


**Figure 3: Prior and posterior distributions of error parameters, when using the range-based *P* and *E* prior error model with different GRACE storage data sets: CSR Spherical Harmonic (SH) and JPL Mascon (MAS) solutions.**

Figure 4 compares prior (GRACE data) and posterior estimates of the basin's water storage *S*. The prior estimates in the top

panel are obtained by predicting water storage from the water balance using the original, uncalibrated data sets for *P*, *E*, *C*, and *Q*. The different permutations in which these data can be combined results in 15 predicted water storage time series in Fig. 4a. We see that the GRACE data closely follow the seasonal dynamics of the prior predicted storages from water balance data. This conveys that both timeseries are approximately in phase but markedly different in magnitude and trend, due to bias and noise in each data set. As shown next, water balance data fusion combines all the data and corrects data errors to yield a closing

water balance in each month.

The second panel of Fig. 4 depicts the calibrated storages, i.e., posteriors shown in green, after fusing all sources of information available on this variable. These posteriors combine three noisy and biased sources of information for storage *S* in a given month: the GRACE observation (if available for that month), the water balance constraint for the current month, and the water balance constraint for the next month (since *S* is initial storage for the next month). Fusing all information together results in

a narrower storage posterior (with less uncertainty) compared to the individual sources.





In this process, the introduced water balance constraints can themselves significantly contribute to reducing the posterior uncertainty of the inferred storage by inducing correlations between variables. This occurs because (noisily) observing $S$ in this constraint linearly links all the a priori independent variables together. Such a relation is encoded in the coefficients of the variables in the water balance equation. For example, to maintain this physical relation, that is, $S$ remains equal to $S_0 + \bar{P} -$

$\bar{E} - Q + C$, the water balance variables with opposite signs must covary together, and vice versa. Accounting for positive correlations between the variables of opposite signs and negative correlations between the variables with the same signs in the water balance can significantly reduce the variance of $S$. The effect of these posterior cross-correlations between water balance variables is shown in Fig. 4b. Note the smaller uncertainty in the final posterior estimates (green band), which account for posterior cross-correlations between the variables, compared to posterior samples (in grey) that do not account for these

posterior cross-correlations (i.e., by independently sampling from the marginal posteriors of $\bar{P}$, $\bar{E}$, $Q$, and $C$).

**Figure 4: Monthly timeseries of GRACE-JPL Mascon observations and storage anomalies predicted from the water balance: (a) using 15 unique uncalibrated combinations of the input variables ($S_0$, $P$, $E$, $Q$, $C$), and (b) using posterior (calibrated) estimates of the input variables ($S_0$, $P$, $E$, $Q$, $C$) with (green band) and without (gray lines) accounting for posterior cross-correlation between the**

**input variables.**



## 5.2 Grid-scale posteriors

While grid-scale posteriors of $P$ and $E$ are obtained in each month, we present here detailed results for two months, which were chosen based on previous analysis (Mourad et al., 2024), namely (a) May 2009, representing a dry month before the monsoon with the highest air temperature and with significant differences between evaporation data products, and (b) July 2009, representing the peak rainy month with large differences between precipitation data products. To guide the interpretation of the spatially distributed posteriors, we appended the land use map (see fig. A1).

The evaporation maps in Fig. 5 show that the posterior mean ($mE^*$) follows the evaporation estimates available from the diverse range of the original gridded data products. For example, in May, the smallest spatial difference between the posterior evaporation mean and the observed evaporation over the irrigated croplands can be seen by SSEBop, with spatial locations for these land use elements having values higher or lower than the posterior mean by $\pm$ 25 mm month$^{-1}$ (Fig. C4 in Appendix C). Over non-crop lands (urban areas and water bodies), the posterior mean has a better agreement with ALEXI and CMRSET, with differences ranging between $-25$ and 25 mm month$^{-1}$. In July, SSEBop displayed the closest agreement with the posterior mean over almost all land use elements. The posterior mean also follows eeSEBAL observed evaporation estimates over irrigated croplands in the lower parts of the basin for this particular month. It is reassuring to see that the generated grid-scale posterior ensemble mean preserves the spatial patterns inherent in the data while combining evaporation information from all products across the different land use elements. This makes sense since the posterior mean is modeled as a weighted average of the minimum and maximum estimates at each grid cell, so that spatial patterns in the original data are maintained. In parallel, the Kalman gain in Eq. (31) distributes the basin-wide constraints to the individual $E$ pixels by explicitly accounting for spatial autocorrelations encoded in the grid-scale covariance matrix ($V_{E_t}$). The results in Fig. 5 also indicate that the posterior uncertainty ($sE^*$) is smaller in the monsoon month (July) than in the dry month (May). This is expected as the data are in better agreement in the peak rainy month than in the dry month. Generally, posterior uncertainty is larger in areas with higher evaporation values (irrigated crop-lands), while non-cropland has both lower evaporation and smaller posterior uncertainty. As such, it can be safely concluded that the spatially distributed calibrated evaporation provides reasonable estimates with acceptable uncertainties that vary with the evaporation magnitude estimated at each pixel.





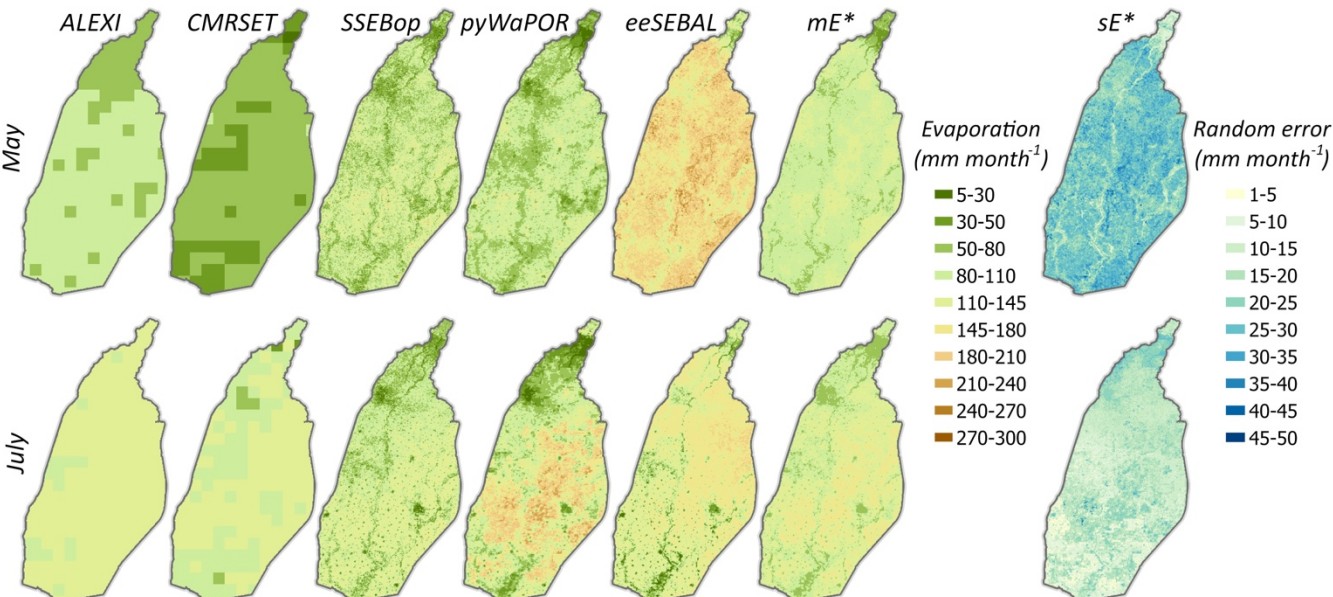

**Figure 5: Prior observations and estimated posteriors of grid-scale evaporation $E$ before the monsoon (May 2009) and for a peak rainy month (July 2009); m$E$\* and s$E$\* denote the posterior mean and standard deviation of evaporation.**

In Fig. 6, we see that the precipitation posterior mean falls between the gridded precipitation data products. The spatial difference maps between the grid-scale observations and the posterior mean show that TRMM observes lower precipitation (by −60 to −35 mm month$^{-1}$), while MSWEP observes higher precipitation (by 25 to 48 mm month$^{-1}$) than the posterior mean. On the other hand, GPM deviates by −21 to 60 mm month$^{-1}$ from the posterior mean (Fig. C5 in Appendix C). As we have seen with the evaporation standard errors, the grid-scale posterior standard errors of precipitation are the highest at spatial locations where the difference across the products is the largest (upper part of the basin), and vice versa.

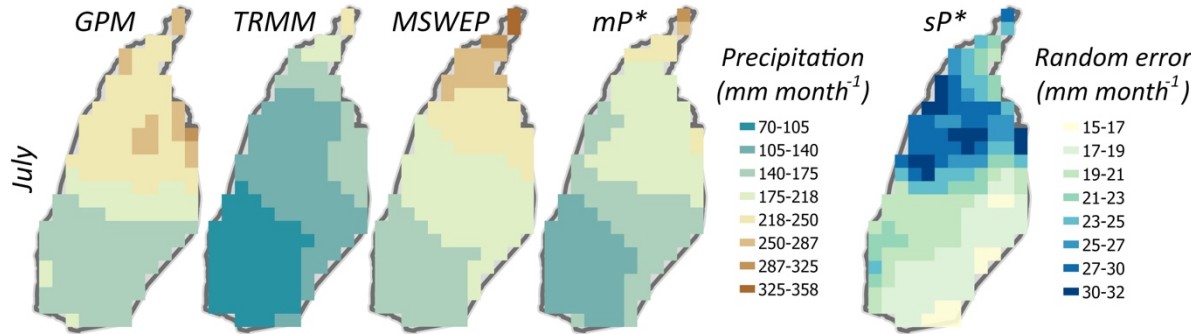

**Figure 6: Prior observations and estimated posteriors of grid-scale precipitation $P$ for the peak rainy month (July 2009); m$P$\* and s$P$\* denote the posterior mean and standard deviation of precipitation.**

Via Eq. (30), the posterior cross-correlation between water variables at the basin scale is propagated to the grid scale, and thus results in posterior cross-correlation between grid-scale $P$ and $E$ as well. The effect of this can be illustrated by computing





derived variables that involve both $P$ and $E$. An example is given in Fig. 7, which shows maps of the posterior mean and standard deviation of $P - E$ for July 2009, the latter with and without posterior cross-correlation between $P$ and $E$. Due to positive posterior cross-correlation between $P$ and $E$ (since they have opposite signs in the water balance), the posterior standard deviation of $P - E$ decreases when accounting for this correlation, but the effect in Fig. 7 is relatively small, and is

observed at some localized spots with the highest reduction in errors of up to 2.5 mm month$^{-1}$. These spots correspond to the locations at which both $P$ and $E$ covary the most, with the cross-covariance subtracted during the computation of the $P - E$ posterior variance. The reason for this relatively small effect is that the water balance constraints introduce other posterior cross-correlations as well, both at the basin scale (Table 2) and at the grid scale via nonlocal posterior cross-correlations between $P$ and $E$ induced by Eq. (30) (i.e., in the posterior, $P$ in a grid cell becomes correlated with $E$ in all other grid cells).

To evaluate the effect of ignoring the posterior correlation between $P$ and $E$, we calculated the distribution of $Q$ from the water balance given the posterior of all other variables for three cases: (i) ignoring all posterior correlation, (ii) ignoring only correlation between $P$ and $E$, (iii) accounting for all correlation. We compare these distributions to a fourth scenario (iv) in which $Q$ is obtained from uncalibrated water balance data. Writing the water balance (Eq. (1)) in terms of $Q$ results in 15 uncalibrated $Q$ estimates generated from the unique combinations of the water balance data. The distribution for this case is

characterized in terms of mean (average across the 15 uncalibrated values) and uncertainty (set as the spread across these values). The results in Fig. 8 indicate that neglecting the posterior cross-correlation between $P$ and $E$ and between all water balance variables, can substantially lead to higher uncertainties (wider distributions), compared to when cross-correlations between all variables are accounted for (narrow distribution). At the extreme end, the $Q$ predicted from the uncalibrated water balance data demonstrated the highest level of uncertainty compared to all other cases, with physically inconsistent values

(negative) and a mean value greatly deviating from the observed $Q$ value. We repeat this experiment for all months and report the performance of the estimated $Q$ from the different approaches compared to the observed Q estimates. The low RMSE value of the calibrated $Q$ posterior mean (0.7 mm month$^{-1}$) compared to $Q_{wb}$ obtained from uncalibrated water balance data (74 mm month$^{-1}$) underscore the importance of (i) water balance data fusion for bias-correcting the original data products and (ii) accounting for posterior correlation, when water balance variables ($P$, $E$, $C$, $S$) are to be used for computing $Q$.

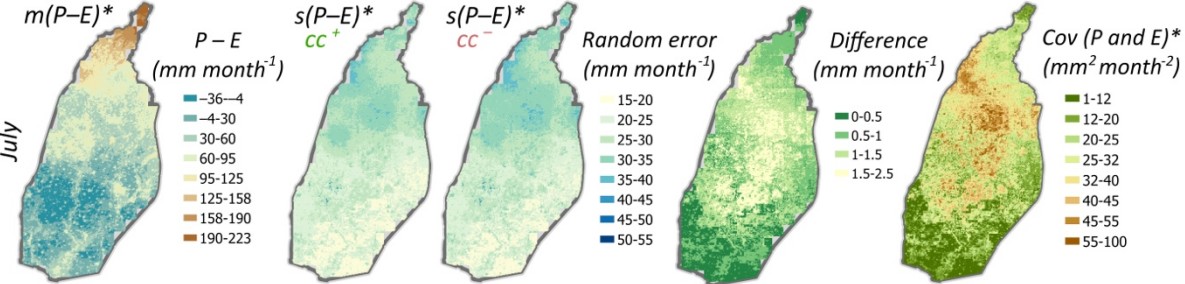


**Figure 7: Grid-scale posterior mean $m^*_{P-E}$ and standard deviation $s^*_{P-E}$ of $P - E$, and $P$ and $E$ covariance for July 2009. The posterior standard deviations are computed for the two cases: with ($cc^+$) and without ($cc^-$) accounting for posterior cross-correlations (cc) between $P$ and $E$, while the difference $cc^- - cc^+$ between these two cases illustrates the impact of accounting for cross-correlations.**




**Table 2: Basin-scale posterior covariance matrix of all water balance variables for July 2009, the diagonal elements represent the posterior variances (mm month$^{-1}$), while the off-diagonal entries represent the posterior covariances between the variables (mm$^2$ month$^{-2}$).**

|  | $S_{t-1}$ | $\bar{P}$ | $\bar{E}$ | $Q$ | $C$ |
|---|---|---|---|---|---|
| $S_{t-1}$ | **562** | *-255* | *62* | 5 | -9 |
| $\bar{P}$ | *-255* | **499** | *49* | 4 | -7 |
| $\bar{E}$ | *62* | *49* | **158** | -1 | 2 |
| $Q$ | 5 | 4 | -1 | **14** | 0.1 |
| $C$ | -9 | -7 | 2 | 0.1 | **23** |

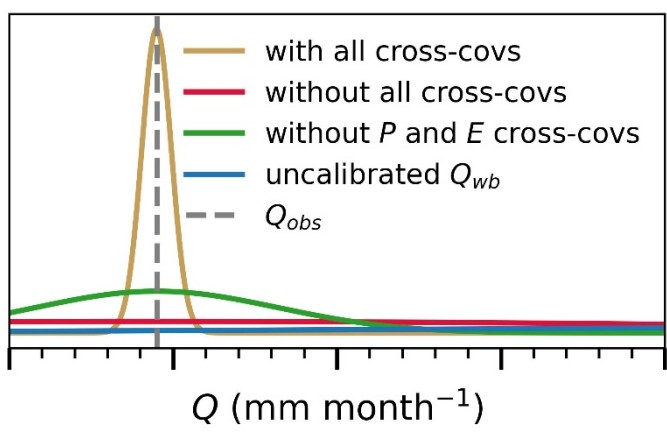


**Figure 8: Distribution of $Q$ in comparison to the observed $Q_{obs}$ value (July 2009), for the following cases: (i) ignoring and accounting for posterior correlations between all water balance variables, (ii) ignoring only posterior correlation between $P$ and $E$, and (iii) $Q_{wb}$ obtained from uncalibrated water balance data.**

## 6 Discussion

This section evaluates some of the modeling decisions and their impact on the results, specifically the choice of GRACE data set and the choice of spatial correlation lengths of $P$ and $E$.

### 6.1 Water balance data fusion with different GRACE data

The Hindon basin studied here is experiencing groundwater depletion (Alam and Umar, 2013). Storage data from GRACE satellite has been used to estimate human-driven water storage changes, such as those caused by intensive irrigation. This data

is either based on spherical harmonic coefficients or mascon basis function, which entails testing the sensitivity of the water balance closure to the chosen GRACE input. The baseline results presented in Sect. 5 are based on GRACE-JPL Mascon (MAS) solution. Here, we compare these closure results to those obtained using the CSR spherical harmonic (SH) solution.





Figure 9 compares the posterior distributions for storage and error parameters from both data sets. The corresponding error parameter posteriors are compared in Fig. 3. The two products show negative depleting trends with widely varying magnitudes
of $-43.3$ mm year$^{-1}$ from the SH and $-88.7$ mm year$^{-1}$ from MAS. Looking at their corresponding posterior distributions, we can see that they both largely follow the GRACE data but with an increase in the seasonal amplitude compared to the data. This is reflected in the inferred $A$ parameter (row 2 in Fig. 3), with an estimated value of 90-100 mm for both cases. The storage posteriors, thus, restore the seasonal signal amplitude, which tends to be more severely attenuated in small-sized basins like the Hindon basin. Additionally, the other storage error parameters (phase $\delta$ and noise $\sigma_S$) distributions are well constrained
compared to their vague prior distributions. A relatively small phase error (time lag) is obtained using the two data sets. However, the JPL-MAS data yields a smaller noise (53 mm) than the CSR-SH (60 mm) data. The fact that these error parameters are time-invariant and that the posteriors are computed using data from all months via iterative smoothing allows the probabilistic water balance model to fill in the data gaps present in GRACE data sets from 2011 onwards.

Figure 3 also depicts the posterior distribution of the precipitation and evaporation error parameters for the two scenarios. The
water balance data fusion run with the CSR spherical harmonic (SH) solution gives more weight to the lower precipitation and evaporation estimates. This is translated into the $w_p$ value of 0.15. Despite that, the posterior evaporation mean lay between the data limits ($w_E$=0.5), the bias scaling factor ($f_E$) masked out its effect (with estimated values smaller than 1), pushing the $E$ posterior towards the lower evaporation end (i.e., the baseline evaporation limit). Figure C6 in Appendix C contains the detailed posterior plots for the water balance variables using the GRACE-SH data sets. Apparently, the GRACE-SH data set's
storage dynamics don't match the other water balance variables in terms of water balance closure, at least with the applied $P$ and $E$ range-based error models. To consistently compare the data fit from both solutions, we normalize the likelihood by the number of observations in each data set. Consequently, the water balance model with GRACE-MAS data has a slightly larger likelihood ($-5.76$) than that with GRACE-SH (likelihood $-6.0$).

This analysis suggests that long-term groundwater depletion in the basin is possibly better captured by the (more severely
declining) mascon data, which has important ramifications for the sustainability of irrigation practices in the basin. This conclusion, however, requires further verification with groundwater level trends from piezometer data.



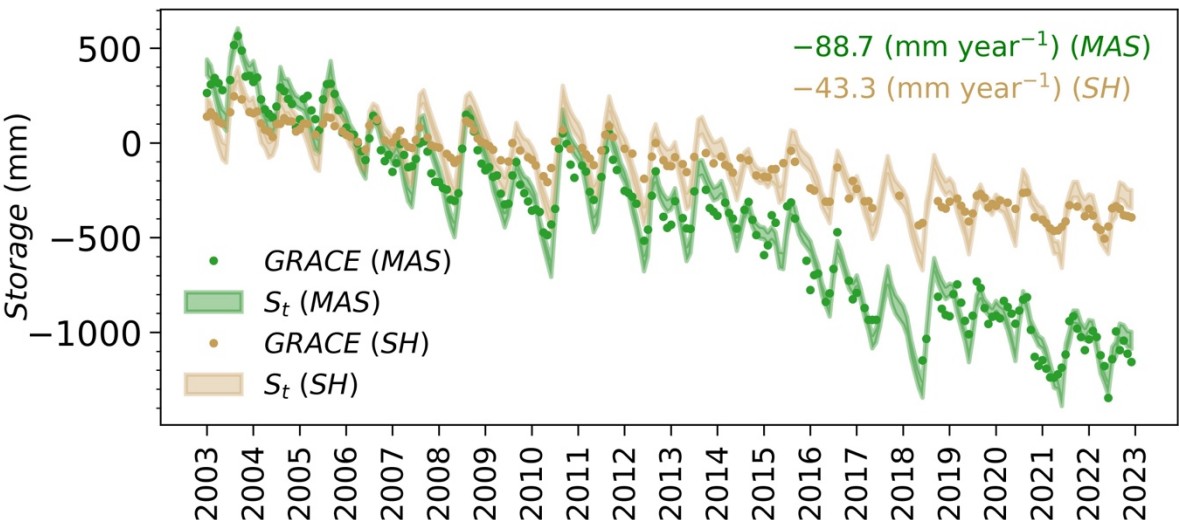

**Figure 9: Storage posteriors for two different GRACE data sets: GRACE-JPL Mascon (MAS) and CSR spherical harmonic (SH) solution.**

## 6.2 Sensitivity to the assumed spatial correlation length-scale parameters

The posterior results in the preceding section are obtained by running the water balance data fusion with fixed Matérn parameters for both the evaporation and precipitation processes. Given the differences in spatial resolution of the analyzed data, the smoothness and correlation length-scale $l_s$ were specified with mid-range values of 1.5 and 50 km, respectively. To evaluate the sensitivity of the results to the chosen value for $l_s$, we set the value of precipitation $l_s$ at 50 km and vary that of evaporation, and vice versa. Table 3 lists these prior conditions, along with the likelihood and average posterior standard deviation for each water balance variable. The results show stability of the posteriors and the likelihood values under all prior settings, except for the cases where precipitation and evaporation $l_s$ are low (e.g., 20 km). In these cases, the probabilistic water balance exhibits the poorest fit to the data (i.e., small likelihood values), and the basin-scale posterior uncertainty of either evaporation or precipitation tends to be underestimated. On the other hand, using mid-range to larger spatial correlation length scale values produces slightly different but substantively comparable posteriors and likelihood values. The relative insensitivity to $l_s$ suggests its value could potentially be chosen to a very large value to speed up the computations: in the limit of perfect spatial correlation, the formulas linking grid and basin scales simplify, which can result in much faster computations for large grids.

**Table 3: Average posterior standard deviation (mm month$^{-1}$) for each water balance variable and likelihood values under different spatial correlation length-scale ($l_s$) prior settings.**

| Spatial correlation length-scale ($l_s$) prior settings (km) | | Likelihood | Average posterior standard deviation (mm month$^{-1}$) | | | | |
|---|---|---|---|---|---|---|---|
| $E$ | $P$ | | $S$ | $P$ | $E$ | $Q$ | $C$ |




| 20 | 50 | −1207.58 | 20.1 | 6.7 | 6.7 | 1.8 | 4.4 |
| 30 | 50 | −1198.62 | 20.5 | 6.7 | 8.4 | 1.8 | 4.5 |
| 40 | 50 | −1193.68 | 20.8 | 6.7 | 9.4 | 1.8 | 4.5 |
| **_50_** | **_50_** | **_−1191.66_** | **_21.3_** | **_6.8_** | **_10.2_** | **_1.8_** | **_4.4_** |
| 60 | 50 | −1245.32 | 24.8 | 6.4 | 10.6 | 1.8 | 4.5 |
| 70 | 50 | −1188.22 | 21.3 | 6.5 | 10.7 | 1.8 | 4.5 |
| 80 | 50 | −1190.69 | 21.1 | 5.8 | 11.1 | 1.8 | 4.5 |
| 90 | 50 | −1190.37 | 21.6 | 6.0 | 11.5 | 1.8 | 4.5 |

**6.3 Sensitivity of the posteriors to the prior canal water import estimates**

The baseline results presented in Sect. 5.1 are based on the canal water import data generated using the approach described in Appendix A (Sect. A2). Here, we evaluate the sensitivity of the probabilistic water balance model to the different approaches used to generate canal water estimates from little available data, compared to the one adopted in the baseline run (we refer to it here as scenario 1). To do so, we include two additional scenarios, one in which the prior canal estimates are computed as the canal design capacities multiplied by their operational time obtained from irrigation schedules (scenario 2), and the other is based on the assumption that the total flows into the basin are 20% of the monthly flows in the UGC main canal (scenario 3). For this evaluation, we assess the model fit given the data (likelihood value) and the average posterior standard deviation (mm month$^{-1}$) for each water balance variable. The resulting water balance variables posteriors showed minimal variations with the different water supplies prior estimates (Table 4). Although the fit is better with scenarios 2 and 3, the posterior error for this term is the lowest for scenario 1. Generally, this analysis shows an insensitivity of water balance posteriors to the canal water data used, and indicates that the prior estimates presented in Sect. 5.1 are sufficient for the goal of constraining the other water balance variables.

**Table 4: Average posterior standard deviation (mm month$^{-1}$) for each water balance variable and likelihood values with different canal water import prior estimates.**

| scenario | Likelihood | Average posterior standard deviation (mm month$^{-1}$) | | | | |
| --- | --- | --- | --- | --- | --- | --- |
| | | *S* | *P* | *E* | *Q* | *C* |
| 1 | **_−1191.66_** | **_21.3_** | **_6.8_** | **_10.2_** | **_1.8_** | **_4.4_** |
| 2 | −1174.55 | 20.7 | 6.4 | 9.8 | 1.8 | 5.3 |
| 3 | −1171.91 | 21.0 | 6.6 | 10.0 | 1.8 | 5.2 |





## 7 Conclusions

This paper presents a multi-scale monthly probabilistic water balance data fusion model for calibrating estimates of each component of the water balance. A key contribution of this paper is the calibration of gridded precipitation and evaporation

estimates in an ensemble approach that fully exploits the prior information content of the data. To achieve this, the introduced methodology is applied at two scales: the basin scale and the grid scale. First, we formulate prior grid-scale and ensemble-based error models for precipitation and evaporation with unknown error parameters that describe the bias and random errors in their spatial fields. The spatially averaged inputs to the precipitation and evaporation error models, along with in situ data (river discharge and canal imports) and storage data from GRACE, then drive a basin-scale probabilistic water balance closure

approach. In this approach, the water balance is treated as a Bayesian model by assigning prior distributions with unknown bias and noise parameters to each water balance variable. Combining these prior distributions with monthly water balance constraints results in posterior estimates of all parameters and variables that jointly close the water balance at the basin scale. The resulting basin-scale precipitation and evaporation posteriors and their cross-correlations are then used to update the prior grid-scale data from the first step. This is attained using a Kalman smoothing algorithm that ensures consistency between the

grid-scale and the basin-scale estimates; that is, spatially averaging the calibrated gridded estimates yields monthly precipitation and evaporation values that jointly satisfy the water balance at the basin scale together with the other variables (C, Q, S).

We apply the introduced methodology to the Hindon Basin, a tributary of the Yamuna River that suffers from unsustainable irrigation practices relying on the local groundwater and imported surface water. The basin-scale results demonstrate that

introducing an independent set of in situ data on surface water imports and river discharge, along with monthly water balance constraints, updates the prior information with new information, automatically adjusts different information sources for each water balance variable, while maintaining a closed water balance. The output of this study highlights the potential of the monthly water balance constraints in substantially reducing uncertainties by accounting for cross-correlations between all water balance variables. In addition, the model yields basin-wide posterior estimates of: (a) error parameters that are well

constrained by all water balance data, and (b) consistent basin-scale water balance variables that jointly close the water balance. Transferring these basin-scale constraints to the grid scale results in: (a) posterior ensemble mean of precipitation and evaporation that fuses the pixel-wise information from extreme prior data bounds (min-max) and finds a balance between the two according to their relative errors, and (b) posterior random errors that reflect the model confidence about the location of the posterior mean at each grid cell.

Explicit to the introduced approach are the assumptions made about the data to characterize their associated errors in a parametric form. For instance, a range-based error model is used to quantify errors in the gridded precipitation and evaporation data. Here, the bias error is described as a weighted average of the minimum and maximum estimates at each spatial location. Whereas, the random error is quantified by scaling the maximum potential error (i.e., scaling one-quarter of the grid-scale full range). As an extension to this analysis, alternative data error models might be worth considering. E.g., a possible alternative



is to use Gaussian mixture models that calibrate each data set individually, while still exploiting the water balance as a constraint.

As for the errors in a single GRACE data set, these are modeled using a noisy sine wave error model with unknown phase and amplitude error parameters that are used to correct for the dominant biases in the GRACE data (i.e., leakages). The sensitivity of the model results to the input data is assessed using two GRACE data sets: spherical harmonic (GRACE-CSR) and mascon

solution (GRACE-JPL). This is done quantitatively by comparing the values of the inferred noise parameters (a smaller value is preferred) and that of the likelihood, as a larger likelihood corresponds to a better fit. While the GRACE-JPL demonstrated a better performance than GRACE-CSR based on these evaluation criteria, this could also be influenced by the assumptions surrounding the precipitation and evaporation error models. Ground-based piezometer data may also help resolve the difference between the downward trends in the two GRACE data sets for the studied basin, which is important for evaluating the

sustainability of irrigation practices in the basin.

Finally, the random errors in the monthly river discharge and canal water import data are more or less fixed to relative errors. A larger prior uncertainty is assigned to the gap-filled canal import data (25%) than to the river discharge (10%). Regardless of the approach used to generate prior canal water import estimates from little available data, these were sufficient to constrain the other water balance terms. Additionally, the generated water balance estimates are not strongly affected by the assumed

errors for the canal water and river discharge data, as these two variables are relatively small compared to the other monthly water balance variables (especially during rainy months when the two processes are driven by precipitation). We also examined the impact of fixing the spatial correlation length scale parameter on model fit and the posterior standard errors. The posterior results showed robustness and model fit improvement across mid-to-large values of prior parameter sets. This suggests that this parameter does not influence the model results in a large way.

A key aspect of the proposed methodology is that it is data-driven, exploiting the water balance constraint as an independent source of information. Therefore, the resulting calibrated data generated here can be used as baseline data sets for multiple applications, such as the validation or calibration of hydrological models, climate studies, and water accounting assessments. Nonetheless, updating the calibrated and spatially distributed estimates of precipitation and evaporation, and storage data generated herein with constraints describing the physical relations between detailed water balance variables (e.g., vertical and

horizontal flows), along with additional in situ data (e.g., groundwater storage observations from piezometers), hold a great premise for synergetic and complementary use of all information sources. This will be the scope of a follow-up study for the investigated basin that aims to constrain water balance stocks and flows and their errors within a water accounting tool that enforces grid-scale water balances (Hessels et al., 2022).

**Code and data availability**

All software used to collect the data is available at https://doi-org.tudelft.idm.oclc.org/10.5281/zenodo.11148992. The source code developed for this research is available via http://doi.org.tudelft.idm.oclc.org/10.5281/zenodo.4116451.



**Author contributions**

RM: methodology, software, formal analysis, validation, visualisation, writing original draft, writing (review & editing). GS: conceptualization, methodology, software, formal analysis, validation, writing (review & editing), supervision, funding

acquisition. VR: data provision, review. WB: funding acquisition, review.

**Competing interests**

The contact author has declared that none of the authors has any competing interests.

**Acknowledgments**

The authors would like to thank Bas Mohrmann for coordinating and facilitating access to the river discharge and canal water

import data used in this study. Also, the authors thank Ankur Rohel, Dikshant Bodana, Vivek Tiwari, and M. Sai Bargav Reddy
       for helping with canal water data access and entry.

**Financial support**

This work is supported by the Dutch Research Council, Government of The Netherlands, through the Meridian Fund (project no. 482.20.303) and by the Department of Science & Technology, Government of India (project no.

DST/TMDEWO/WTI/NWO/CGAW/2020/14), under the jointly funded Cleaning the Ganga and Agri-Water initiative and the
       Hindon Roots Sensing (HIROS) project: River Rejuvenation through Scalable Water- and Solute Balance Modelling and
       Informed Farmers' Actions.

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



**Appendix A**

The following subsections summarize the available information on the Hindon basin land use elements (Fig. A1), share of
canal water imports in the water balance (Fig. A2), canal network (Figs. A3 and A4), its operation, irrigation canal delivery
data, and processing steps.

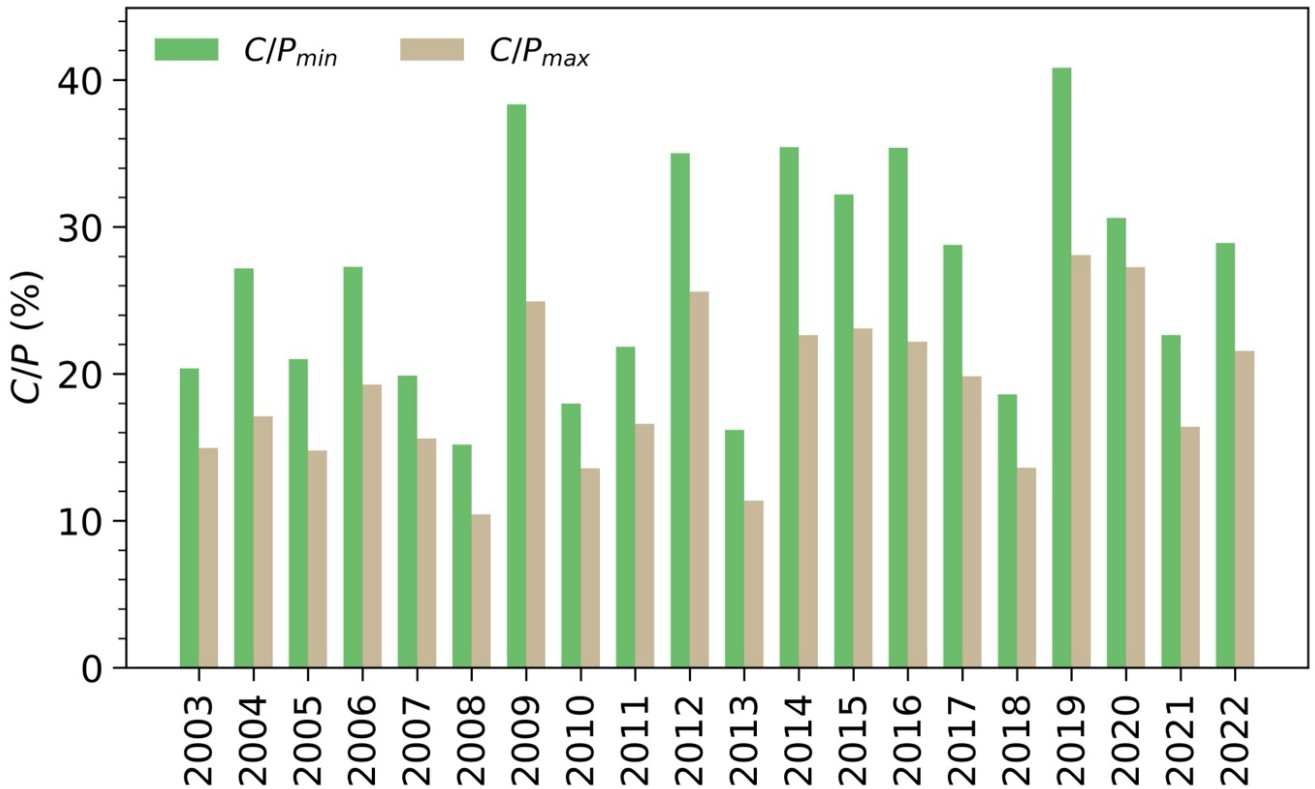

**Fig A1: Share of the canal water imports in the water balance, expressed as the percentage of the minimum and maximum remote**
**sensing-based precipitation observations.**





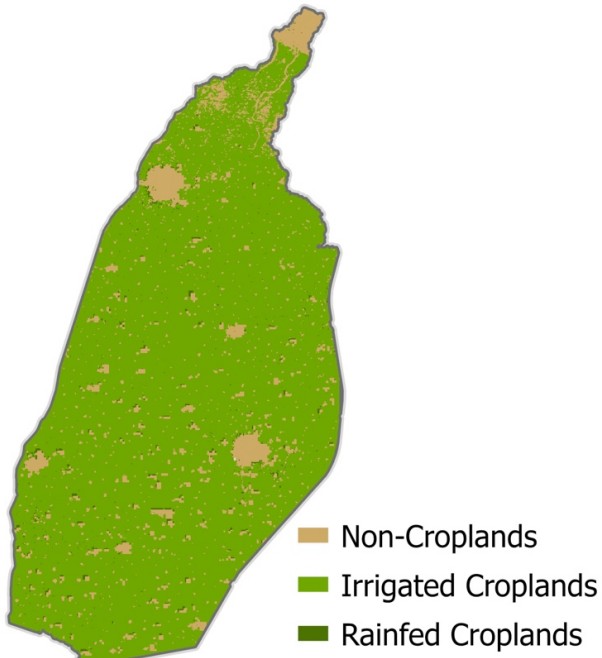

**Fig A2: Land use elements distribution in the Hindon basin from Gumma et al. (2022).**

## A1. Description of the canal irrigation system and its operation

Figs A3 and A4 provide an overview of the canal network's spatial distribution and design capacities. While these main canals run continuously, their distributaries are operated in rotation (Ahmad, 1991; Chaube et al., 2023). The canal distributaries in the Hindon basin are operated on an 'on-off' basis as shown in Table A1. The irrigation schedules (also known as the rosters) are prepared for two irrigational seasons:

    (1) Kharif season in which the main irrigated crops are sugarcane, rice, cotton, maize, vegetables, and fodder crops
(2) Rabi season in which the principal crops are wheat, barley, mustard, and peas.

**Table A1: Example of seasonal operational schedule (the ON-OFF basis) for Eastern Yamuna Canal (EYC) and Upper Ganga Canal (UGC) distributaries. Kharif spans from March to September 2021, while Rabi covers Oct 2021 to March 2022.**

| Main Canal | Intake | 3 weeks ON- 1 week OFF | 4 weeks ON | 2 weeks ON- 2 weeks OFF | 1 week ON- 3 weeks OFF |
|---|---|---|---|---|---|
| UGC (Kharif) | Deoband Branch | April,May,July | June,Aug,Sept | | |
| | Tanshipur | July,Aug | June,Sept | April | May |
| | Mohammadpur | July, Aug | June,Sept | April,May | |
| | Mansurpur | July,Aug | June,Sept | April,May | |



| | | | | |
|---|---|---|---|---|
| UGC (Rabi) | Deoband Branch | Feb | Nov,Dec,Jan | Oct |
| | Tanshipur | | Nov,Dec,Jan | Feb,Oct |
| | Mohammadpur | | Dec,Jan | Feb,Oct,Nov |
| | Mansurpur | Dec | Nov, Jan,Feb | Oct |
| EYC (Kharif) | Nagla | Aug | July | Apr,May,June |
| | Babail | Aug | July | Apr,May,June |
| | Sarakadi | Aug | July | Apr,May,June,Sep |
| | Megh Chappar | | June,July,Aug | Apr,Sep |
| | Chidbana | | June,July,Aug | Apr,Sep |
| | Nalhera | Aug | June | Apr,July,Sep |
| | Redi | Aug | June | Apr,July,Sep |
| | Rampuri | Aug | June | Apr,May,July,Sep |
| | Kallarpur | | July,Aug | Apr,May,June,Sep |
| EYC (Rabi) | Nagla | | | Oct,Nov,Dec,Jan,Feb |
| | Babail | | | Oct,Nov,Dec,Feb |
| | Sarakadi | | | Oct,Nov,Dec,Mar |
| | Megh Chappar | | | Oct,Nov,Jan,Mar |
| | Chidbana | | | Oct,Nov,Jan,Mar |
| | Nalhera | | | Oct,Nov,Jan,Mar |
| | Redi | | | Oct,Nov,Jan,Mar |
| | Rampuri | | | Oct,Dec,Jan,Mar |
| | Kallarpur | | | Oct,Dec,Feb |






**Fig A3: A schematic of the UGC main canal and its distributaries with their respective discharge capacities (m³ s⁻¹).**





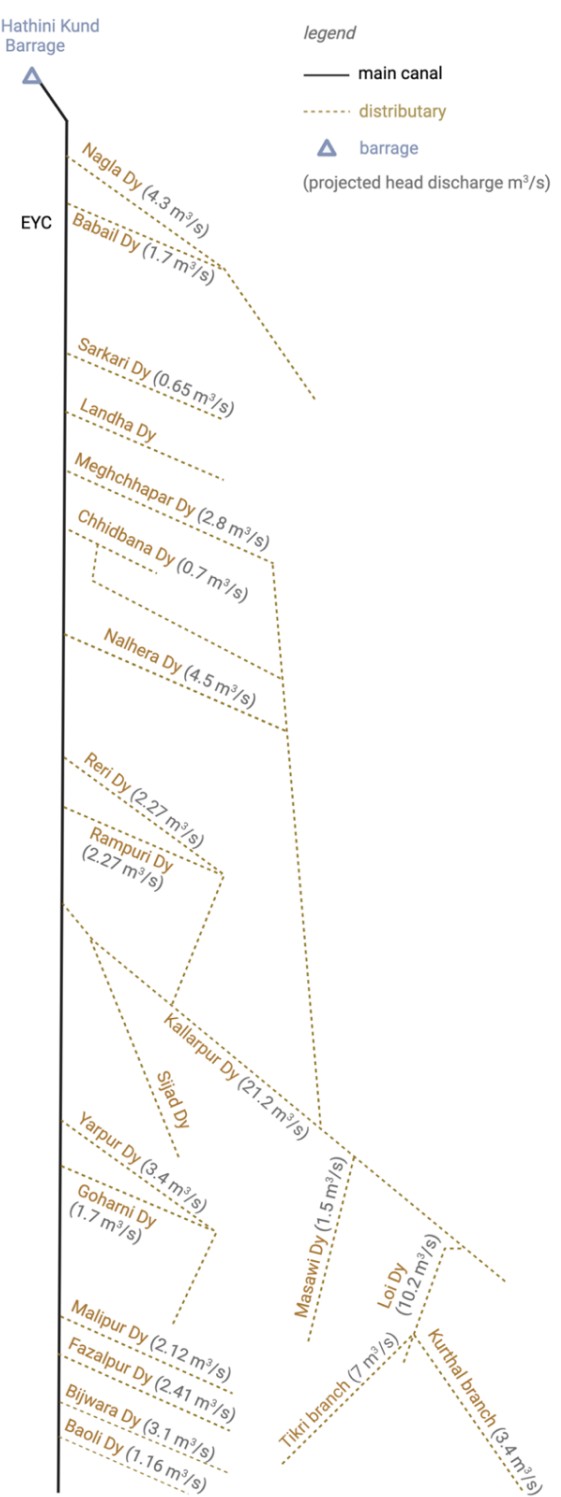

**Fig A4: A schematic of the EYC main canal and its distributaries with their respective discharge capacities (m³ s⁻¹).**



## A2. Description of irrigation canal delivery data and processing steps

Constrained by the limited availability of data on water deliveries, we use an extrapolation approach to fill in the data gaps. The approach integrates two data sources, the irrigation schedules and the actual flow measurements from in-situ data, to produce a complete monthly timeseries. Explicit in this method is the assumption that the canal operations do not vary widely between years in which the conditions are similar. For instance, if years "A", "B," and "C" are classified as "dry years," the canal operations don't differ much across these years. In other words, if year "A" has unknown estimates, these can be estimated from year "B" or the mean of the three years, depending on the number of years with canal data. Therefore, each group of distributaries is filled out based on two criteria: the data source and the number of years with available data. For the canals, such as the Deoband parallel, with actual flow measurements and ≥ 5 years of data availability, we solely rely on these data for extrapolation. To this end, we distinguish the years into three conditions (wet, dry, and normal years) using the Palmer Drought Severity Index (PDSI) Terra climate data set as a proxy. Then, every unknown year is filled with the long-term average of the years falling under the same category. For example, when a year such as 2015 is identified as "normal", this year is filled with the mean of the canal delivery observed across all other "normal" years. A similar procedure was applied to the EYC lower intakes with actual flows from 2018 to 2022 to fill in missing data for similar years outside this period. For the other intakes with only a few years of data, however, we filled them differently. For instance, the lower intake points of the UGC with flow measurements for only two years, 2021 (dry) and 2022 (wet), we use these years as benchmarks for filling similar wet or dry years. For "normal" years on these intakes, we use the irrigation schedules instead. Explicitly, in cases with only a single-year irrigation schedule, the flows are obtained by multiplying the full design capacity by the operational time. This step produces conservative and monthly variable flow estimates due to the operational time variation of each distributary. Since the Deoband branch contributes a significant share of the total canal water supplies into the basin, we fill its missing data using the irrigation schedules to avoid the impacts of assuming no variation in operation across the years. In cases where single-year data is available for distributaries such as Bijwara and Baoli, we use the actual flows for this year (2018) as a reference to fill all other years. As for distributaries without data from design or actual flows, we assume these might not add much to the overall imports into the Hindon basin. At this stage, the canal delivery data is at the daily timestep. The last step is to aggregate to a monthly timescale, sum all intakes, and normalize by the basin area to obtain the gap-filled canal water imports expressed in water depth units (mm month$^{-1}$).



**Table A2: Available information on the distributaries and their coverage. The data source column distinguishes between actual (measured) and planned (scheduled) irrigation canal delivery data.**

| Main Canal | Distributaries | data source | data availability |
|---|---|---|---|
| EYC | Nagla | weekly irrigation schedules | 2021-2022 |
| | Babail | weekly irrigation schedules | 2021-2022 |
| | Sarkari | weekly irrigation schedules | 2021-2022 |
| | Landha | not available | not available |
| | Meghchhapar | weekly irrigation schedules | 2021-2022 |
| | Chhidbana | weekly irrigation schedules | 2021-2022 |
| | Nalhera | weekly irrigation schedules | 2021-2022 |
| | Reri | weekly irrigation schedules | 2021-2022 |
| | Rampuri | weekly irrigation schedules | 2021-2022 |
| | Kallarpur | Daily actual flows | 2018-2023 |
| | Sijad | not available | not available |
| | Olra Escape | not available | not available |
| | Yarpur | Daily actual flows | 2018-2023 |
| | Gohari | Daily actual flows | 2018-2023 |
| | Malipur | Daily actual flows | 2018-2023 |
| | Fazalpur | Daily actual flows | 2018-2023 |
| | Bijwara | Daily actual flows | 2018 |
| | Baoli | Daily actual flows | 2018 |
| UGC | Deoband Branch | Daily actual flows | 2013-2014-2016 |
| | Parallel Deoband | Daily actual flows | 2018 to 2022 and 2013-2014-2016 |
| | Tanshipur | weekly irrigation schedules | 2021-2022 |
| | R. Mohammadpur | weekly irrigation schedules | 2021-2022 |
| | R.Jauli | Daily actual flows | 2021-2022 |
| | Mansurpur | Daily actual flows | 2021-2022 |
| | Khatauli Escape | Daily actual flows | 2021-2022 |
| | R. Salawa | Daily actual flows | 2021-2022 |



**Appendix B**

**B1. A major challenge arising from the use of uncalibrated remote sensing water balance data**

The following figure shows that imbalance errors due to combining uncalibrated remote sensing products for each water balance variable can be significant ($\sim \pm 200$ (mm month$^{-1}$)). This is related to errors associated with the data. Therefore, there is a need to quantify and reduce these uncertainties.

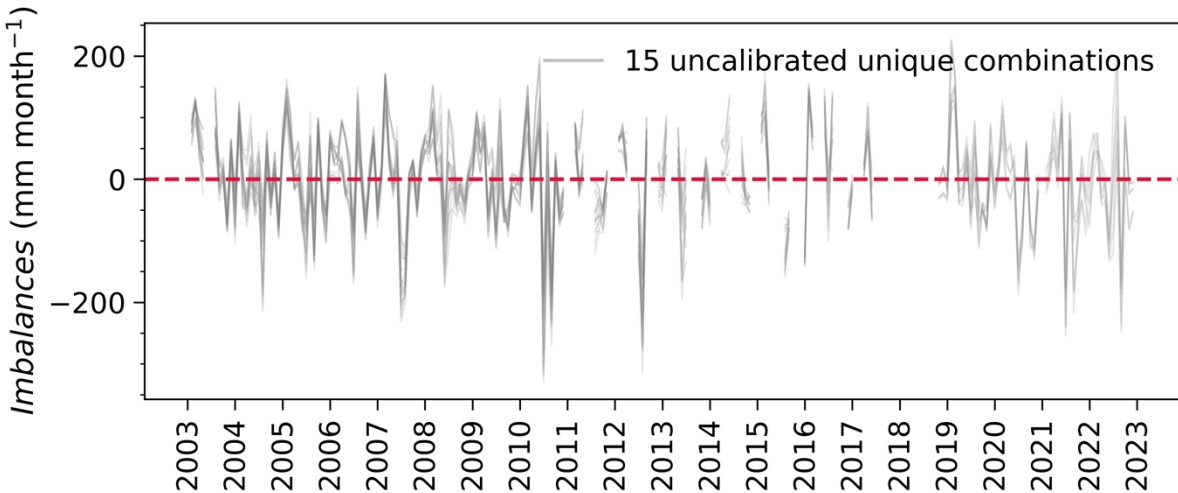

**Fig B1: Monthly timeseries of the imbalances (water balance errors) for each of the 15 unique uncalibrated combinations of the variables.**

**B2. Computing posteriors of individual variables, parameters, and posterior cross-correlations between water balance variables**

This section briefly explains how all posteriors are computed. For the probabilistic water balance model described in section 4.1 of the main text, we can represent its joint distribution as $p(x, \theta, S_{obs})$. This distribution consists of two parts: the monthly water balance variables x (comprising 5N + 1 variables including $S_0$, $\bar{P}_t$, $\bar{E}_t$, $Q_t$, $C_t$ and $S_t$), where N is the number of months and is equal to 240 for 20 years of data considered in this paper, $S_0$ is the initial basin water storage at the start of the first month and $S_{obs}$ is the storage timeseries, and the parameter vector θ, including $w_P$, $r_P$, $w_E$, $f_E$, $r_E$, $a_Q$, $b_Q$, $a_C$, $b_C$, $\sigma_S$, $A$, and $\delta$. Our objective is to compute the full posterior distribution, which integrates all available information sources (Gaussian distributions) of θ and x.

The posterior of the parameter vector is expressed in the following form:

$$p(\theta|S_{obs}) \propto p(S_{obs}|\theta)p(\theta) \tag{B1}$$

where $p(\theta)$ is the prior distribution for the parameters and corresponds to the product of the individual parameter priors defined in the previous basin-wide error model section, while the term $p(S_{obs}|\theta)$ refers to the parameter likelihood function. This function scores each set of bias parameters (e.g., $w_E$, $f_E$) and noise parameters (e.g., $r_E$, $\sigma_S$). A large likelihood for the





parameters is one that shifts the storage predicted from the water balance closer to the storage observations. As we show below, this likelihood also connects the DE-MCMC for parameter sampling and EP for posterior computations of water balance variables.

In the basin-wide inference setup, an outer MCMC loop iteratively proposes sets of parameter candidates (samples) using a non-parametric proposal (jumping) mechanism (Ter Braak and Vrugt, 2008). For each set of sampled parameters by MCMC, the EP (message passing) algorithm operates in an inner loop, computing the (a) unnormalized posterior density of the parameters proposed by MCMC (Eq. (B1)) and the conditional water balance posteriors $p(x \mid S_{obs}, \theta) = \frac{p(x, S_{obs}|\theta)}{p(S_{obs} |\theta)}$. The normalizing constant of the conditional water balance posterior ($p(S_{obs} |\theta)$) is the likelihood in Eq. (B1). EP evaluates the

posterior odds of the model outputs $p(x \mid S_{obs}, \theta)$ given the sampled parameters, guiding the DE-MCMC decisions in accepting or rejecting new parameters. Compared to the standard single forward-backward algorithm, EP approximates the exact posterior distributions with Gaussian distributions sharing matching moments (mean and variance). It involves multiple back-and-forward passes over the data using the entire timeseries until all posteriors stabilize. A key aspect of the EP algorithm is its ability to efficiently handle non-Gaussian posteriors, which arise from physical non-negativity constraints we force on

water balance variables ($\bar{P}, \bar{E}, Q, C$). Typically, a small number of iterations are sufficient to achieve convergence due to the mild non-Gaussianity induced by these constraints. The EP algorithm yields conditional water balance posteriors (conditioned on the $S_{obs}$ data and the parameter vector θ). Instead of the conditional posterior distribution, we are interested in the marginal posterior distribution $p(x \mid S_{obs})$ over the individual water balance variables ($S_0, \bar{P}, \bar{E}, Q$, and $C$). Such distributions are obtained by integrating (~averaging) $p(x \mid S_{obs}, \theta)$ over the parameter posterior distribution $p(\theta|S_{obs})$, effectively accounting

for the parameter uncertainty in the final water balance estimates:

$$p(x \mid S_{obs}) = \int p(x \mid S_{obs}, \theta) \, p(\theta|S_{obs})d\theta \tag{B2}$$

This paragraph explains how to compute posterior cross-covariances between the different water balance variables given the posterior marginal distributions of each variable (as output by the water balance data fusion model). For this, we will need two building blocks to construct the joint precision matrix (Θ) equal to the inverse of the covariance matrix ($V^*$), namely: the

variables' linear relation through the water balance coefficients $\varphi = [1, 1, -1, -1, 1]$ extracted from the water balance constraint at each time step: $S = S_0 + \bar{P} - \bar{E} - Q + C$ and "uncorrelated priors" of each variable. The term "priors" reflects uncertainty before considering the relation of each variable with other variables. With the marginal distributions available at this point, we can back-calculate these priors such that they satisfy the marginals through a message-passing algorithm. This algorithm relies on decomposing the marginal precisions ($\tau$) for each variable into: (a) prior precisions (messages) for each

variable and (b) precisions (messages) sent to each variable from all other variables via the water balance. Specifically, in each month, we have two conditions: the water balance constraint and the probabilistic constraints in the form of marginal Gaussian distributions written separately for individual variables $x$ as $p(x) \sim \mathcal{N}(m_x^*, v_x^*)$, with posterior mean and variance obtained



from Eq. (27) in the main text. Expressed in terms of precisions, we can write the decomposed posterior precision of S ($\tau_S$), for example, as: $\tau_S = \tau_{S\rightarrow} + \tau_{\rightarrow S}$ where $\tau_{S\rightarrow}$ is the prior precision from S, and $\tau_{\rightarrow S}$ is the precision to S from the water balance.

The latter can be obtained from all other variables by propagating uncertainty through the linear water balance equation. As such, the relationship between the posterior and prior variances of S that fulfill the two defined conditions becomes:

$$\frac{1}{v_S^*} = \frac{1}{v_{S\rightarrow}} + \frac{1}{v_{S_0\rightarrow}+v_{\bar{P}\rightarrow}+v_{\bar{E}\rightarrow}+v_{Q\rightarrow}+v_{C\rightarrow}} \tag{B3}$$

We can repeat the above step for all other variables to create a linear system of six equations with six unknowns (prior variances) that we need to solve for:

$$\begin{cases} \frac{1}{v_S^*} - \frac{1}{v_{S\rightarrow}} - \frac{1}{v_{S_0\rightarrow}+v_{\bar{P}\rightarrow}+v_{\bar{E}\rightarrow}+v_{Q\rightarrow}+v_{C\rightarrow}} = 0 \\ \frac{1}{v_{S_0}^*} - \frac{1}{v_{S_0\rightarrow}} - \frac{1}{v_{S\rightarrow}+v_{\bar{P}\rightarrow}+v_{\bar{E}\rightarrow}+v_{Q\rightarrow}+v_{C\rightarrow}} = 0 \\ \frac{1}{v_{\bar{P}}^*} - \frac{1}{v_{\bar{P}\rightarrow}} - \frac{1}{v_{S_0\rightarrow}+v_{\rightarrow S}+v_{\bar{E}\rightarrow}+v_{Q\rightarrow}+v_{C\rightarrow}} = 0 \\ \frac{1}{v_{\bar{E}}^*} - \frac{1}{v_{\bar{E}\rightarrow}} - \frac{1}{v_{S_0\rightarrow}+v_{\bar{P}\rightarrow}+v_{\rightarrow S}+v_{Q\rightarrow}+v_{C\rightarrow}} = 0 \\ \frac{1}{v_Q^*} - \frac{1}{v_{Q\rightarrow}} - \frac{1}{v_{S_0\rightarrow}+v_{\bar{P}\rightarrow}+v_{\bar{E}\rightarrow}+v_{\rightarrow S}+v_{C\rightarrow}} = 0 \\ \frac{1}{v_C^*} - \frac{1}{v_{C\rightarrow}} - \frac{1}{v_{S_0\rightarrow}+v_{\bar{P}\rightarrow}+v_{\bar{E}\rightarrow}+v_{Q\rightarrow}+v_{\rightarrow S}} = 0 \end{cases} \tag{B4}$$

Where $v_S^*, v_{S_0}^*, v_{\bar{P}}^*, v_{\bar{E}}^*, v_Q^*, v_C^*$ are the posterior marginal variances of the variables: Storage $S$, initial storage state for the first

month $S_0$, $\bar{P}$, $\bar{E}$, $Q$, and $C$, respectively. While, $v_{S\rightarrow}, v_{S_0\rightarrow}, v_{\bar{P}\rightarrow}, v_{\bar{E}\rightarrow}, v_{Q\rightarrow}$, and $v_{C\rightarrow}$ are their corresponding prior variances.

The resulting priors are used to generate the joint precision matrix $\Theta$ of S0, P, E, Q, and C, which combines two terms:

$$\Theta = \varphi\varphi'\tau_{S\rightarrow} + diag([\tau_{S_0} \text{ or } \tau_{S_{t-1}\rightarrow}, \tau_{\bar{P}\rightarrow}, \tau_{\bar{E}\rightarrow}, \tau_{Q\rightarrow}, \tau_{C\rightarrow}]) \tag{B7}$$

The first term represents a matrix that encodes how each variable varies with every other variable in the water balance. This term is scaled by the prior precision from S ($\tau_{S\rightarrow}$), whereas the second term is a diagonal matrix of the other individual water balance variables prior precisions. The inverse of the joint precision matrix gives the posterior covariance matrix between all

variables (in a given month), which contains all posterior cross-correlations.

## Appendix C

### C1. Water balance priors and posteriors

This section displays the results of the seasonal and annual posterior means of the water balance variables compared to their corresponding prior data.



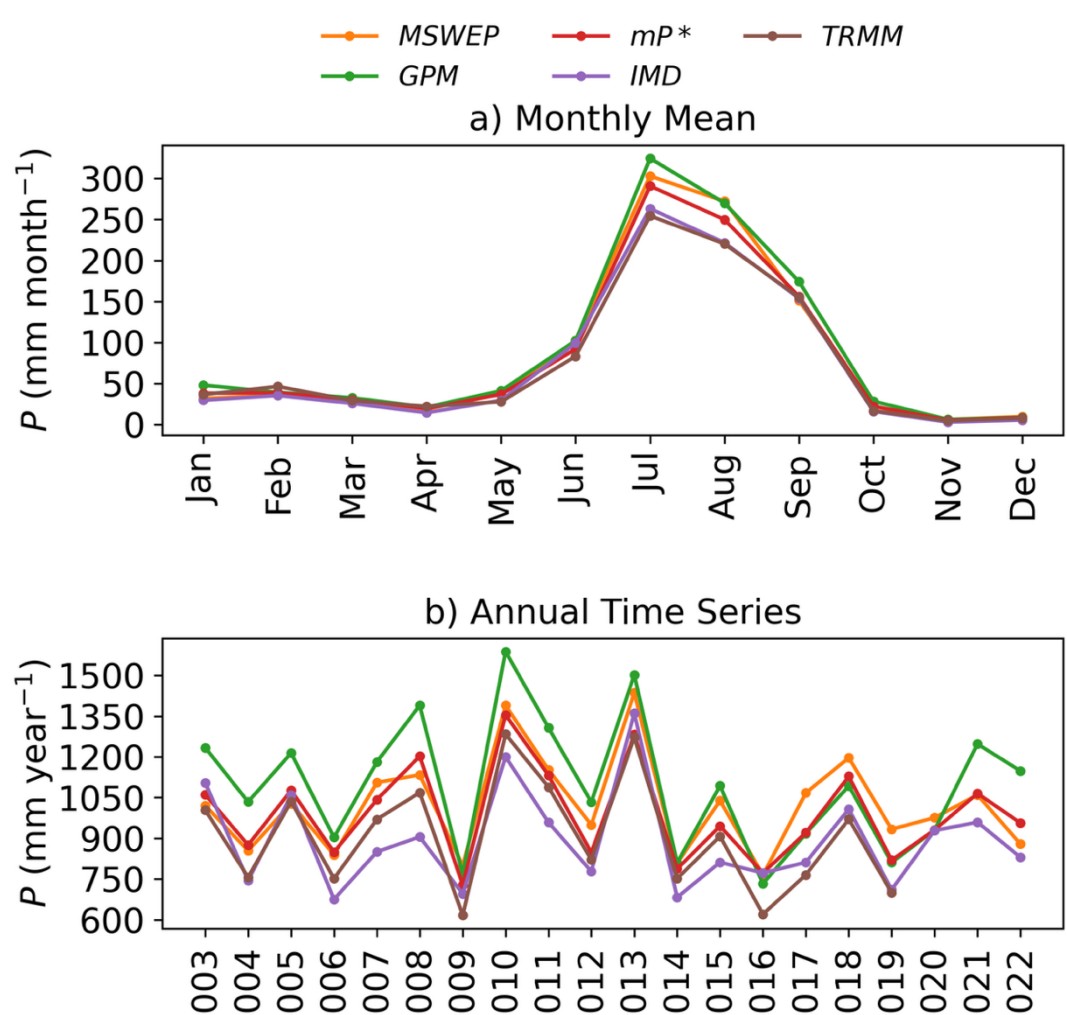


**Fig C1: Comparison of posterior mean (mP\*) to the prior data sets describing the precipitation (*P*) variable in the Hindon basin, the top panel depicts (a) monthly mean P, and the bottom panel (b) shows the annual P time series.**





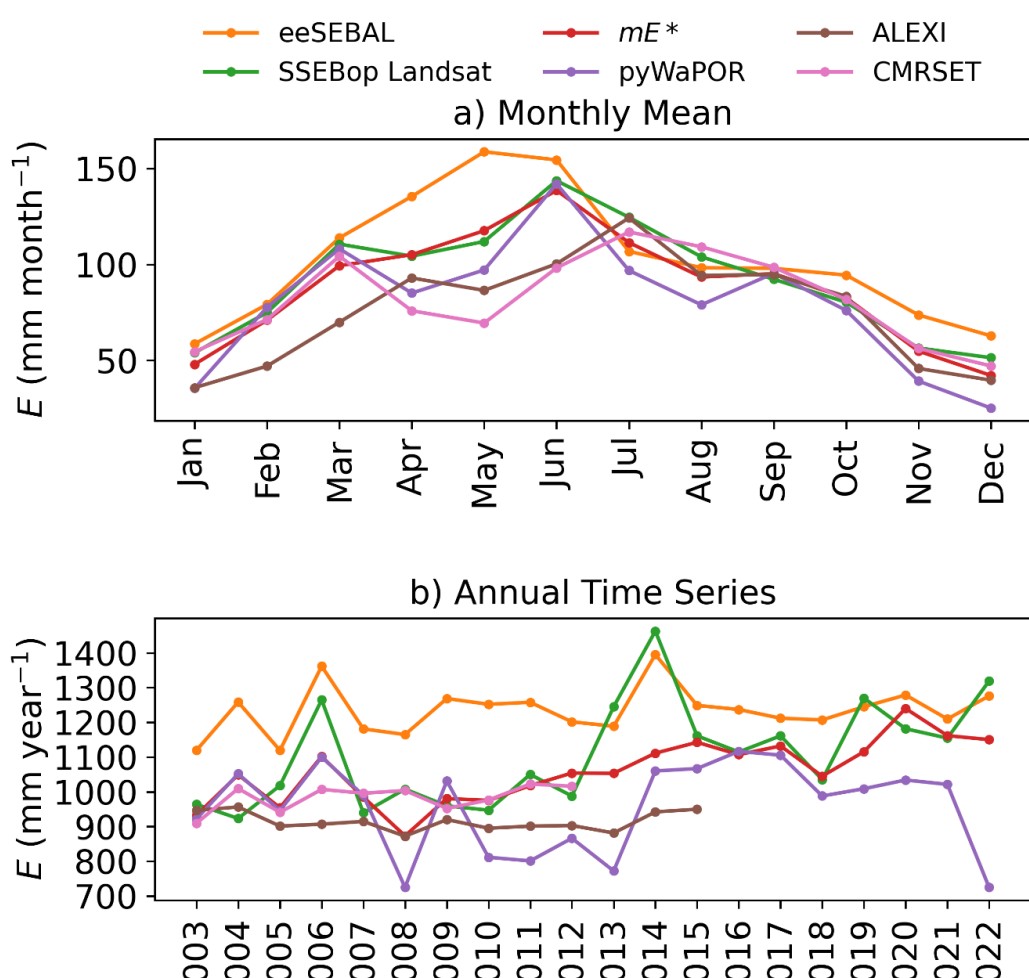

**990**    **Fig C2: Comparison of the posterior mean (m*E*\*) to the prior data sets describing the evaporation (*E*) variable in the Hindon basin, the top panel depicts (a) monthly mean *E*, and the bottom panel (b) shows the annual *E* time series.**





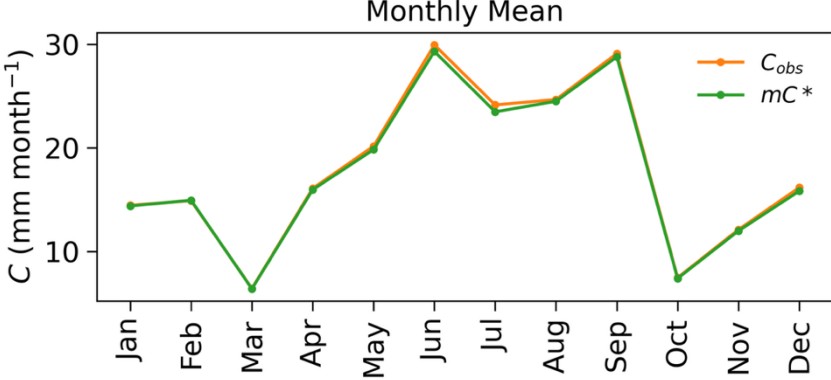


**Fig C3: Comparison of the monthly mean posterior mean (m$C$\*) to the prior data describing the surface water imports ($C$) variable in the Hindon basin.**

## C2. Comparison of individual gridded products to the posterior means

This section presents the spatial maps of the deviation of the individual precipitation and evaporation products from their

posterior mean.

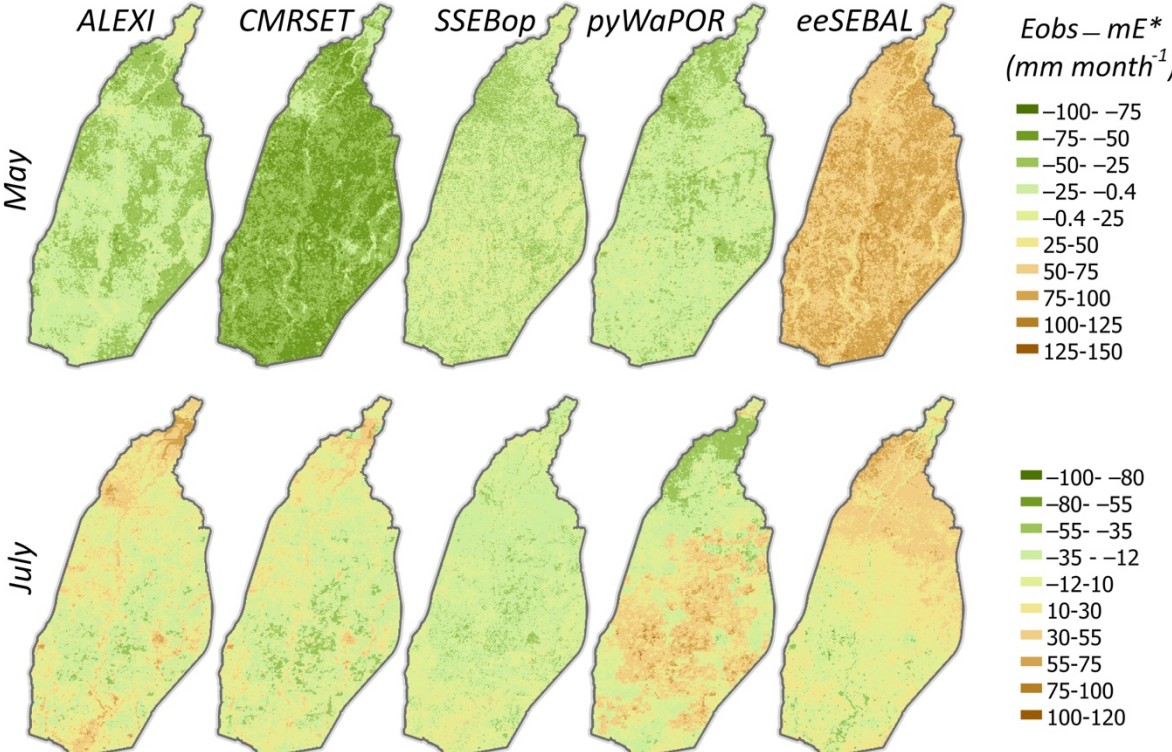

**Fig C4: Spatial deviation of the evaporation ($E$) observations from the posterior mean ($E_{obs} - mE^*$) for May and July of 2009.**



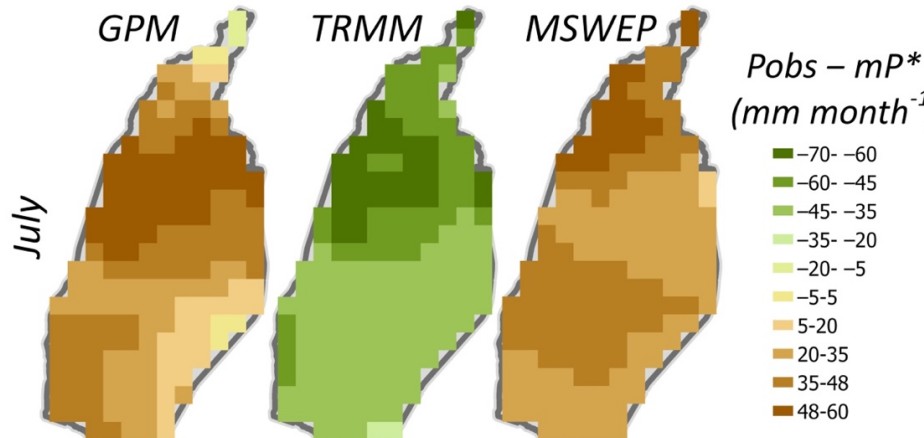

**Fig C5: Spatial deviation of the precipitation ($P$) observations from the posterior mean ($P_{obs} - mP^*$) for May and July of 2009.**





## C3. The sensitivity of the posterior results to the GRACE input data



**Fig C6:** Posterior and prior water balance plots correspond to the case using the GRACE CSR spherical harmonic (SH) solution as an input to the probabilistic water balance model.
