# Peer review of "Joint calibration of multi-scale hydrological data sets using probabilistic water balance data fusion: methodology and application to the irrigated Hindon River Basin, India"

_EGUsphere, 2025_

## Author Comment (AC1)

Reviewer #1:

"This manuscript presents a novel probabilistic water balance data fusion approach for calibrating multi-scale hydrological datasets. The methodology is innovative, addressing the challenge of reducing uncertainties in datasets by integrating them through water balance constraints. The approach provides a framework for both basin-scale and pixel-scale applications. The application to the Hindon River Basin demonstrates practical utility, with reasonable error estimates and clear improvements in data consistency. The paper is well-written, structured, and accessible, making a substantial contribution to water resource management and hydrological modeling. However, some areas, such as the clarity of methodological details and validation against independent data, could be strengthened to enhance the robustness and reproducibility of the findings. Suggestions are as follows:"

We would like to thank the referee for the time and effort reviewing our manuscript, and for the valuable feedback received. We are pleased that the reviewer found the paper well written and appreciate the recognition of the novelty of the presented methods. In the following, we address the reviewer's detailed comments.

Reviewer comment 1:
"In Section 2, beginning on line 117, you describe the Hindon Basin and the separation of two irrigation seasons (Kharif and Rabi), yet it is unclear if the rotated crops use the same land or if they are in adjacent regions. It would be helpful to add a sentence or two clarifying this."

Reply on comment 1:
We will add the following sentence at the end of section 2 to mention that the distributaries take off from main canals to serve fixed command areas irrigated year-round, with crops rotated between Kharif and Rabi crops:
"Irrigation water is diverted from the canals to supply the basin through off-takes that serve fixed command areas with crops rotated between Rabi or Kharif crops."

Reviewer comment 2:
In your results, the validation could be strengthened. Are you able to compare your estimates against any in-situ records? Reported standard errors are useful, but which component dominates the uncertainty (precip, evaporation, storage, discharge, canal imports)? Standard errors are provided but there is no discussion of comparisons with independent ground-truth data or other datasets not used in calibration. Including such validation would enhance confidence in the results.

Reviewer comment 3:
Discussion would benefit from a short explanation on generalization. For example, can this approach work in snow dominated or urban catchments or is it basin specific?

Reply on comments 2 and 3:
Indeed, it will be useful to report the order of the standard errors of the different water balance variables, and it might be more interesting to highlight this at the start of the paper. Therefore, we will edit the abstract to reflect the order:

"An application to the irrigated Hindon River basin in India illustrates that the approach generates physically reasonable estimates of all basin-scale variables, with average standard errors decreasing in the following order: 21 mm month$^{-1}$ for storage, 10 mm month$^{-1}$ for

evaporation,7 mm month$^{-1}$ for precipitation, 4 mm month$^{-1}$ for irrigation canal water imports, and 2 mm month$^{-1}$ for river discharge".

We also agree that it's good to reflect on model generalizability and validation; therefore, we will add the following separate section:

"6.4. Extensions

The presented methodology is motivated by the availability of diverse water balance remote sensing data, with very few in situ data available for the Hindon basin, making independent validation of our estimates challenging. For this reason, we evaluated our results using soft validation techniques. For example, for evaporation, in the absence of in situ data, we compared the evaporation posterior estimates with reference evapotranspiration and local irrigation practices (see Sect. 5.1). As for the precipitation posterior estimates, in Sect. 5.1 and Appendix C, these were compared to the spatially interpolated rain gauge dataset for the basin from the Indian Meteorological Department (IMD), keeping in mind the potential underestimation of precipitation by this dataset (Goteti and Famiglietti, 2024). Evaluating the total water storage estimates with independent data is more challenging. In a follow-up study, we will introduce separate rootzone and groundwater balance constraints, with the aim of estimating their contributions to the total water balance. At that point, it will become possible to use available remote sensing soil moisture data, as well as in-situ groundwater level data, for evaluation. Furthermore, adding these additional constraints and data will allow for updating the posterior estimates in this paper.

 Additionally, the presented methodology is general and can be applied to other gauged river basins. For example, an application to multiple semi-arid basins was reported in Schoups and Nasseri (2021). The method has several advantages that allow this, including the straightforward and flexible implementation, as it consists of two separate parts: an error model specification that can be customized to fit specific settings, and the model solver that automatically computes the posterior distributions. In addition, the method is set up to rely on in situ and satellite data that inherently capture the hydrological processes. For example, the precipitation data sets estimate the precipitation phase (snow and rain), while evaporation data sets can be used to differentiate between the different land use classes. However, in snow-dominant basins where precipitation data sets might underrepresent this process, or in urban-dominant settings where coarse-resolution evaporation data products might overestimate evaporation, it may be valuable to tailor the error models to local conditions in order to improve the results. Perhaps, this could be achieved by complementing the precipitation error models with other satellite data sets, like the snow cover, snow depth, or temperature products, for better snow detection and mapping. Moreover, evaporation error models can be complemented with land use maps while also considering the use of high-resolution evaporation products or other data sets with improved evaporation estimates of heterogeneous urban surfaces. The major assumption here is that formulating the error models by exploiting ancillary information would allow it to solve for error parameters and water balance variables under varying climatic zones and settings. Alternatively, for this purpose, we could combine the data error models with hydrological models that explicitly account for detailed processes and differentiate between the different hydrological responses".

Reviewer comment 4:
"Figures with more than one panel (starting with Figure 2) need tags (a, b, c, etc) and the caption should refer to each panel specifically for clarity (like you did for Figure 4). In Table 2, indicate

the meaning of the underlined values. Table 3, Table 4, again indicate the bold and underline importance."

Reply on comment 4:
Thank you for the suggestions regarding figures and tables. We will add the alphabetic labels for the subplots of Figures 2 and 3. The figures and their captions will be modified as follows:

[Figure]

"Figure 2: Monthly water balance posteriors (90% credible intervals in green) for: (a) $P$ (precipitation), (b) $E$ (actual evapotranspiration), and (c) $C$ (canal water imports), and their corresponding observations (dots). Reference evapotranspiration computed here using ERA5 meteorological input variables is shown as ($ET_0$). The overbars of the labels in $P$ and $E$ plots indicate that these values are obtained through spatial averaging of the gridded data sets for each month $t$. Each year's label indicates the start of the year (January)."

[Figure]

"Figure 3: Prior and posterior distributions of error parameters for: (a) evaporation, (b) storage, and (c) precipitation, canal water import, and river discharge when using different GRACE storage data sets: CSR Spherical Harmonic (SH) and JPL Mascon (MAS) solutions."

We will also edit the table 2 caption as follows: "Basin-scale posterior covariance matrix of all water balance variables for July 2009, the diagonal elements represent the posterior variances (mm month$^{-1}$ ) (shown in bold), while the off-diagonal entries represent the posterior covariances between the variables (mm$^2$ month$^{-2}$ ) (underlined values)"

Also, we will add this line to the table 3 and 4 captions to enhance their clarity: "The bold and underlined values represent the results of the baseline setup of the water balance data fusion model, with GRACE-JPL Mascon data set and fixed correlation length parameters at 50 km".

Goteti, G. and Famiglietti, J.: Extent of gross underestimation of precipitation in India, Hydrology and Earth System Sciences Discussions, 2024, 1-40, 2024.
Schoups, G. and Nasseri, M.: GRACEfully closing the water balance: A data-driven probabilistic approach applied to river basins in Iran, Water Resources Research, 57, e2020WR029071, 2021.

---

## Author Comment (AC2)

Reviewer #2:

This work proposes a two-step method, i.e., from basin scale to grid scale, to produce a water budget closed datasets by introducing the Bayesian model with predefined prior distribution and posterior parameter estimation, considering the covariance between water components and the entire time series, under a specific case study in the severely irrigated Hindon River Basin, India. This work tried to introduce an innovative theoretical basis and apply it to a basin with abundant discharge records. Although the logical structure of the work is clear, the theoretical introduction and the equations are hard to follow since there is no deductive process of the equations provided and the code and data are not accessible with the provided links. This makes it hard for the readers to follow the work and estimate the robustness of the work.

We would like to thank the referee for the time and effort reviewing our manuscript, and for the valuable feedback received. We understand that the theoretical introduction of the methods and equations might be hard to follow. We will improve the flow of equations by adding intermediate steps in the equations that more clearly show how one equation leads to the next. Also, we will update the link to the data and software, as follows:

"Code and data availability:
The software used to collect the data is available at https://doi.org/10.5281/zenodo.11148992, and the source code developed for this research is available at http://doi.org/10.5281/zenodo.4116451."

Reviewer  major comment 1:
"the spatial scale problem is not thoroughly discussed. How were the different scales between water components in the budget closure equation handled? Is the resolution 50 km really feasible in such a small basin that is only one pixel wide and two pixel height?"

Reply on major comment 1:

We agree this point deserves more attention, and we will add the following paragraph before Table 1 to reflect on the challenge of the scale difference between the water balance variables:

"As can be seen in Table 1, the native resolution of the different individual data sets and between the water balance variables varies widely, posing a challenge in their merging process. In principle, the water balance data fusion can be performed at any spatial resolution; however, for this study design, we try to preserve the native information as much as possible by choosing to resample all precipitation data sets to a common resolution of 0.05° and all evaporation data sets to a common resolution of 250m. An alternative design would be to resample all water balance variables to the same resolution (e.g., 250m), but this might introduce artefacts, for example, when up-sampling from coarse to high resolution."

As for the 50 km, this refers to the spatial correlation length ($l_s$) that we specify a priori to generate the covariance matrices of precipitation (Eqs. 5-6) and evaporation. The basic assumption here is that the values of the grid cells in the spatial domains of precipitation and evaporation are spatially correlated, i.e., the correlation between the grid values depends on the distance between the grid locations. Grid cells that are close to each other tend to have high correlation. As the spatial separation between grid cells increases, the correlation decreases. Often, information on the correlation length parameter is not available. In such a case, and given the differences in scales between the data sets, we rely on specifying a mid-range value of

the prior correlation length, which is about half the basin's length from North to South (line 238). Finally, we assess the sensitivity of the results to the chosen correlation length in sect 6.2.

Reviewer major comment 2:
"L173/L884-885. About the gap-filling method, the authors made the assumption that the canal operations do not vary widely between years in which the conditions are similar. Is it possible to use the existing data to validate the assumption? I mean compare the data in the years with similar conditions to check whether that assumption is tenable."

Reply on major comment 2:
Unfortunately, we are constrained by the amount of data available, making cross-validation difficult. Specifically, we at most have five years of data, and these years can be distributed between wet, dry, and normal years, leaving us with no out-of-sample data for cross-validation. E.g., if among these 5 years, 2 years are dry, we simply took the average of these two years to fill all other missing dry years. In cases where we know that a distributary might significantly contribute to the overall imports to the basin, e.g., the Deoband branch, we follow a different approach that avoids any average-based related biases and relies on using the design discharge and operation time to estimate the canal water imports for the missing years. In this vein, using the canal design capacities for gap-filling results in conservative (upper bound) estimates of the irrigation water imports, which can also be seen as acceptable approximate initial values for the Bayesian methods used in the study. Additionally, in sect 6, we include scenario 2, where, instead of filling the missing years with the data average of years with similar conditions, we use the upper bound estimates (design capacities multiplied by the operation time). The results show insensitivity to the canal water estimates used, as these are much smaller than the surface exchanges (e.g., precipitation and evaporation). To account for potential uncertainties around all assumptions we make to generate the corresponding full timeseries, we attach a large prior uncertainty (25%) to these gap-filled estimates, i.e., a wide confidence interval around their values.

Reviewer major comment 3:
"It will be much clearer and easier to follow if a framework diagram is provided in the method section."

Reply on major comment 3:
Good suggestion, a flow diagram can aid the reader in grasping the overall flow of the methods section. We will add the following diagram:

[Figure]

Grid-scale priors
Eqs. (2-6) & (12-13)

$$\mathcal{N}\left(\begin{bmatrix} P \\ E \end{bmatrix} \middle| \begin{bmatrix} m_P \\ m_E \end{bmatrix}, \begin{bmatrix} V_P & 0 \\ 0 & V_E \end{bmatrix}\right)$$

Grid to basin-scale averaging
Eq. 25

$$\begin{bmatrix} \bar{P} \\ \bar{E} \end{bmatrix} = \begin{bmatrix} \phi_P' & 0 \\ 0 & \phi_E' \end{bmatrix}\begin{bmatrix} P \\ E \end{bmatrix}$$

Grid-scale update
Sect. 4.2

$P$ $E$

$\bar{P}$ $\bar{E}$

$C$ — $\mathcal{N}(C|m_C, s_C^2)$ Eqs. (18-21)

BWB fusion Sect. 4.1

$\mathcal{N}(S_0|m_{S_0}, s_{S_0}^2)$ — $S_0$ — BWB Eq. (1) — $S$

$\mathcal{N}(Q|m_Q, s_Q^2)$ Eqs. (18-21) — $Q$

$\mathcal{N}(S|m_S, s_S^2)$ Eqs. (22-24)

"Figure caption: a factor graph showing the variables in circles and constraints as squares for a single month. Three constraints are incorporated in the water balance data fusion method, including: a prior Gaussian distribution assigned to each water balance variable (squares attached to each variable), the basin-scale water balance (BWB) constraint that links all water balance variables together, and the spatial averaging constraint that links grid-scale and basin-scale variables for precipitation and evaporation. Computation of posteriors of individual variables proceeds in two steps. The first step is basin-scale water balance data fusion (BWB fusion, see section 4.1). This step involves multiple forward (blue arrows along the edges) and backward (dotted blue arrows) passes over the data using the entire timeseries to compute the posteriors of all water balance variables that jointly close the water balance (Schoups and Nasseri, 2021). The second step computes grid-scale posteriors for precipitation and evaporation from their basin-scale posteriors using a Kalman smoothing algorithm (green arrow, see sect. 4.2)."

Reviewer major comment 4:
"About the matrix variables mentioned in all equations, for example, in Eq. 5, it is better to provide the size parameters of each matrix."

Reviewer major comment 5:
"For me, the relationships between equations are quite independent and the connections are weak. For example, Eq. 3-8, the input and output of each equation are vague thus hard to understand the method itself as a whole. The same issue exists for the entire theoretical part."

Reply on major comments 4 and 5:
We will add the information about the size of the evaporation and precipitation correlation matrices. These are now mentioned in the following modified sections, which have been modified to show additional intermediate steps when going from one equation to the next.

Sect. 3.1.1

The independence structure between $P_t$ and $E_t$ shown in Eq. (2) allows us to represent the errors in each variable separately. For each month $t$, we typically have multiple gridded precipitation products, with unknown bias and random errors. To characterize these errors, the grid-scale precipitation bias and random error models for all elements in the mean vector $m_{P_t}$ and the square root of all diagonal entries of the autocovariance matrix $V_{P_t}$ are defined using Eqs. (3-4), respectively:

$$m_{P_t} = P_t^{obs,min} + w_p\left(P_t^{obs,max} - P_t^{obs,min}\right) \tag{3}$$

$$s_{P_t} = r_P \frac{1}{4}\left(P_t^{obs,max} - P_t^{obs,min}\right) \tag{4}$$

Equation (3) models the systematic bias in the gridded precipitation for each month $t$ by describing the precipitation prior mean $m_{P_t}$ as a function of two parts. The first part represents the baseline precipitation estimated as the minimum across all the products ($P_t^{obs,min}$), while the second consists of a weighted deviation from this base value. Specifically, the latter denotes the full range of the grid-scale observed precipitation ($P_t^{obs,max} - P_t^{obs,min}$) with a weight parameter $w_p$. The bias parameter $w_P$ takes on an unknown value between 0 and 1, with a logit-normal prior distribution of parameter $\mu = 0$ and scale parameter $\sigma = 1.4$ to reflect prior uncertainty about the bias. In other words, $w_P$ controls the relative position of the $m_{P_t}$ within the observed precipitation space (range). As it approaches 0, more weight is given to the minimum across all products ($P_t^{obs,min}$), whereas approaching a value of 1 gives more weight to the maximum across all products ($P_t^{obs,max}$).

Random errors in the precipitation are modeled using Eq. (4). This model expresses the prior gridded standard deviation $s_{P_t}$ for each month $t$ as a function of the maximum potential random error, defined as a quarter of the range: $\frac{1}{4}\left(P_t^{obs,max} - P_t^{obs,min}\right)$. A noise parameter ($r_P$) is used to scale this conservative quantity and is given a quasi-uniform prior distribution between 0 and 1.

To account for the effect of spatial correlation of the random error component (Eq. (4)), we write the precipitation prior covariance matrix $V_{P_t}$ in terms of the grid-scale standard deviations and a grid-scale auto-correlation matrix:

$$V_{P_t} = S_P R_P S_P \tag{5}$$

where $S_P$ is a diagonal matrix containing the grid-scale $s_{P_t}$ values for all locations of the spatial field (Eq. (4)), and $R_P$ is the correlation matrix that captures the spatial dependence structure. $R_P \in \mathbb{R}^{n_P \times n_P}$, where $n_P \times n_P$ is the matrix dimension, and $n_P$ equals 176, representing the total number of grid cell locations of the precipitation spatial domain. $R_P$ is jointly estimated from all precipitation data using an isotropic parametric correlation function with the following form (Handcock and Stein, 1993):

$$C_{\mathcal{M}}(d|l_s, v) = \frac{1}{2^{v-1}\Gamma(v)}\left(\frac{d}{l_s}\right)^v K_v\left(\frac{d}{l_s}\right) \tag{6}$$

where $C_{\mathcal{M}}$ is the Matérn correlation function for variables separated by distance $d$. This correlation model is flexible and widely used, with two functions: gamma function $\Gamma(.)$ and the modified Bessel function $K_v(.)$ (Abramowitz and Stegun, 1968). $C_{\mathcal{M}}$ also consists of two unknown nonnegative parameters, namely the spatial correlation length scale $l_s$ and a spatial smoothness parameter $v$. A value of $v$ approaching 0 indicates a rough spatial process, while the process is smoother when $v$ approaches infinity. Since the smoothness parameter is usually small in many applications (Chen et al., 2022), while it increases as the aggregation time increases (Sun et al., 2015), we choose a balanced value between a rough and smooth random field, i.e., $v$, fixed at 1.5. On the other hand, the correlation length scale ($l_s$) defines an average length scale on which grid cells are correlated with each other. In principle, $l_s$ ranges from 0 (the case of uncorrelated grid cells) and extends to a scale larger than the spatial domain length

(the case of maximally correlated pixels). With no prior information on the $l_s$ parameter, we fix it at 50 km (~1/2 the basin's length from North to South). The sensitivity of the results to the fixed $l_s$ will be evaluated in section 6.2.

Since water balance data fusion (Schoups and Nasseri, 2021) uses basin-scale error models, we derive these from the above-described grid-scale models by spatial averaging. Specifically, the basin-scale prior mean $m_{\bar{P}_t}$, variance $v_{\bar{P}_t}$, and standard deviation $s_{\bar{P}_t}$ in month $t$ follow from Eqs. 3, 4 and 5:

$$m_{\bar{P}_t} = \phi_P' m_{P_t} = \phi_P' \left[ P_t^{obs,min} + w_p \left( P_t^{obs,max} - P_t^{obs,min} \right) \right] \tag{7}$$

$$v_{\bar{P}_t} = \phi_P' V_P \phi_P = \phi_P' S_P R_P S_P \phi_P = \phi_P' (r_P D_P) R_P (r_P D_P) \phi_P \tag{8}$$

$$s_{\bar{P}_t} = \sqrt{v_{\bar{P}_t}} = r_P \sqrt{\phi_P' D_P R_P D_P \phi_P} \tag{9}$$

$$\bar{P}_t \sim \mathcal{N} \left( m_{\bar{P}_t}, s_{\bar{P}_t}^2 \right) \tag{10}$$

$$\bar{P}_t \geq 0 \tag{11}$$

where $\phi_P$ is the spatial averaging operator used to derive basin-scale moments from grid-scale moments (i.e., $n_P \times 1$ vector with each element equal to $1/n_P$, where $n_P$ is the number of spatial locations in the precipitation spatial field). $\phi_P'$ is the transpose of $\phi_P$. We also used $S_P = r_P D_P$, where $D_P$ is a diagonal matrix containing the $\frac{1}{4} \left( P_t^{obs,max} - P_t^{obs,min} \right)$ values (from Eq. 4) for all grid cells within the precipitation spatial domain. All basin-averaged input quantities to Eqs. (7-8) are precomputed from the precipitation data sets, and the constant but unknown parameters $w_p$ and $r_P$ are estimated as part of the water balance data fusion (see Sect. 4.1). Finally, the last two equations in the precipitation error model treat the basin-scale calibrated precipitation $\bar{P}_t$ for each month $t$ as a random draw from a truncated normal distribution. The truncation at zero ensures physical consistency (nonnegative precipitation).

Sect. 3.1.2
As with precipitation, an evaporation error model with the following range-based form is adopted:

$$m_{E_t} = f_E \left[ E_t^{obs,min} + w_E \left( E_t^{obs,max} - E_t^{obs,min} \right) \right] \tag{12}$$

$$s_{E_t} = r_E \frac{1}{4} \left( E_t^{obs,max} - E_t^{obs,min} \right) \tag{13}$$

The bias in evaporation is modeled with two spatial and time-invariant calibration parameters, namely: $w_E$ and $f_E$. The parameter $w_E$ is the weight that interpolates between the monthly gridded evaporation extrema $E_t^{obs,min}$ and $E_t^{obs,max}$ in each month $t$. An additional scaling factor ($f_E$) is incorporated to account for potential bias outside the observed range. On the other hand, Eq. (14) quantifies evaporation prior uncertainty ($\frac{1}{4} \left( E_t^{obs,max} - E_t^{obs,min} \right)$) scaled via the spatially and temporally constant noise parameter ($r_E$). All parameters are treated as random variables with prior distributions. The $w_E$ and $r_E$ parameters are specified with a vague prior distribution bounded between 0 and 1, specifically, flat logit-normal with location parameter $\mu = 0$ and scale parameter $\sigma = 1.4$. Whereas, $f_E$ is given a lognormal prior with mode at 1 (no bias) and a coefficient of variation CV of 50%.

The basin-scale evaporation error models are derived from the grid-based models defined above, using the same spatial averaging process applied to precipitation (Eqs. (7-8)). The averaging formulas are then obtained as follows:

$$m_{\bar{E}_t} = \phi_E' m_{E_t} = f_E \phi_E' \left[ E_t^{obs,min} + w_E \left( E_t^{obs,max} - E_t^{obs,min} \right) \right] \tag{14}$$

$$s_{\bar{E}_t} = r_E \sqrt{\phi_E' D_E R_E D_E \phi_E} \qquad (15)$$

$$\overline{E_t} \sim \mathcal{N}\left(m_{\bar{E}_t}, s_{\bar{E}_t}^2\right) \qquad (16)$$

$$\overline{E_t} \geq 0 \qquad (17)$$

where $D_E$ is a diagonal matrix whose diagonal entries containing the $\frac{1}{4}\left(E_t^{obs,max} - E_t^{obs,min}\right)$ values for all grid cells within the evaporation spatial domain. All inputs of Eqs. (15-16) are precomputed from the evaporation data sets while the unknown parameters $f_E$ and $r_E$ are solved as part of the water balance data fusion (Sect. 4). $\phi_E$ is the spatial averaging operator, and the $R_E$ term stands for the correlation matrix, which captures the spatial dependencies between the evaporation grid cells. $R_E \in \mathbb{R}^{n_E \times n_E}$, where $n_E \times n_E$ is the matrix dimension, and $n_E$ equals 71235, representing the total number of grid cell locations of the evaporation spatial domain. For the large-sized evaporation data sets considered here, we parameterize the evaporation correlation matrix using a Matérn kernel implemented within a stochastic variational Gaussian Process (Hensman et al., 2015) with fixed parameter $l_s$ at 50 km and $\nu$ at 1.5.

Similar to precipitation, the basin-scale calibrated evaporation $\overline{E_t}$ for month $t$ is treated as a random draw from a truncated normal distribution Eqs. (17-18). The truncation at zero ensures physical consistency (nonnegative evaporation).

Reviewer major comment 6:
"L545. The labels and tick marks of x and y missed in Fig. 8".

Reply on major comment 6:
Since the river discharge data are classified (can't be made publicly available), the labels and tick marks of x and y are not shown in Fig. 8. We will update the caption of Fig. 8 as follows:

Figure 8: Distribution of $Q$ in comparison to the observed $Q_{obs}$ value (July 2009), for the following cases: (i) ignoring and accounting for posterior correlations between all water balance variables, (ii) ignoring only posterior correlation between $P$ and $E$, and (iii) $Q_{wb}$ obtained from uncalibrated water balance data. The labels and the tick marks are not shown due to data sharing restrictions.

Reviewer minor comment 1:
"L125 the abbreviation of Central Water Commission (CWC) should be explained near the figure instead later in L168".

Reply on minor comment 1:
That's true; we will define CWC in the Figure 1 caption as follows:

"Figure 1: Location of the Hindon basin in the Uttar Pradesh state of India (inset map); with a detailed view of the basin featuring its boundaries, topographic profile, and the location of the Galeta outlet, where a river discharge station that belongs to the Central Water Commission (CWC) network of India is located. The main map shows the irrigation scheme with reservoirs and canal system. Topographic basemap sources: Esri, USGS, FAO, NPS, GIS user community, and others."

Reviewer minor comment 2:
"L241. the symbols mpt and vpt with Eq.7-8 are different."

Reply on minor comment 2:
We have removed these notations for conciseness and to enhance clarity.

Reviewer minor comment 3:
"L255 I think "Evapotranspiration" is better than "Evaporation" throughout the paper."

Reply on minor comment 3:
At the outset of introducing the probabilistic water balance model (line 130), we define evapotranspiration as evaporation (including transpiration). Later, we use the term "evaporation" throughout the paper for conciseness.

Reviewer minor comment 4:
"L581. There is no ground-water level data? It is weird that the discharge of the canals have been paid great attention while no ground-water data available in such a heavily ground-water-based irrigated basin."

Reply on minor comment 4:
The reviewer is right, the inclusion of groundwater level as an independent evaluation would be valuable; however, in this study, we only considered the total water storage. In a follow-up study, we plan to extend the methodology presented here to incorporate detailed information, such as groundwater pumping, soil moisture, and groundwater level data, where we also focus on separating the rootzone from the groundwater contribution. See also our response to Reviewer 1's comment on validation.

Abramowitz, M. and Stegun, I. A.: Handbook of mathematical functions with formulas, graphs, and mathematical tables, US Government printing office1968.
Chen, H., Ding, L., and Tuo, R.: Kernel packet: An exact and scalable algorithm for Gaussian process regression with Matérn correlations, Journal of machine learning research, 23, 1-32, 2022.
Handcock, M. S. and Stein, M. L.: A Bayesian analysis of kriging, Technometrics, 35, 403-410, 1993.
Hensman, J., Matthews, A., and Ghahramani, Z.: Scalable variational Gaussian process classification, Artificial intelligence and statistics, 351-360,
Schoups, G. and Nasseri, M.: GRACEfully closing the water balance: A data-driven probabilistic approach applied to river basins in Iran, Water Resources Research, 57, e2020WR029071, 2021.
Sun, Y., Bowman, K. P., Genton, M. G., and Tokay, A.: A Matérn model of the spatial covariance structure of point rain rates, Stochastic Environmental Research and Risk Assessment, 29, 411-416, 2015.

---

## Author Comment (AC3)

Reviewer #1

We thank you for the valuable feedback received. The comment on the model output validation using independent data has motivated us to look into additional data, such as weather and crop variables, to further validate our results. We will add figure C4 below, comparing the trend in our estimated evaporation with independent crop yield data, and edit lines 425-432 as follows:

"Finally, we note the interplay between all variables: water balance ($P$, $E$, $C$, and $Q$), weather, and crop variables. Despite relying on external water supplies, the basin has been experiencing a decreasing trend in the river discharge (Dwivedi and Yadav, 2025), coupled with increased evaporation from 2010 to 2022 (see the annual time series of the posterior mean in Fig. C2). The increase in posterior mean evaporation mirrors the increasing trend in the sugarcane yield and leaf area index from 2011 (Fig. C4). This aligns with a period of increasing temperatures (see Fig. C4) and with the period when farmers started adopting new high-yielding sugarcane varieties in the basin, as reported by the Indian Council of Agricultural Research. The accelerating rate of evaporation in our estimates follows a similar trend in the Landsat-based SSEBop evaporation data set (Fig. C2). This shows, on the one hand, the usefulness of the water balance constraints in evaluating the data products, and on the other hand, that simplified ET models like SSEBop can sufficiently reproduce the evaporation dynamics.
The interplay between variables is also pronounced in dry years such as 2009 and 2016, showing a decline in the posterior mean precipitation, together with the canal water imports. The limited availability of rainfall and surface water for irrigation during droughts, along with rising temperatures in recent years (Fig. C4), does not, however, depress the evaporation rates because crop water demand is met by increased groundwater pumping. The effect of unsustainable groundwater pumping is particularly notable in the dynamics and trend of basin water storage, which we discuss next."

And the first lines of section 6.4 become:

"The presented methodology is motivated by the availability of diverse water balance remote sensing data, with very few in situ data available for the Hindon basin, making independent validation of our estimates challenging. For this reason, we evaluated our results using soft validation techniques. For example, for evaporation, in the absence of in situ data, we compared the trend in our estimated evaporation with independent crop yield data, and the seasonal dynamics with known cropping and irrigation patterns (see Sect. 5.1)."

[Figure]

Figure C4. Interplay between the annual: evaporation posterior mean ($mE^*$), air temperature ($T_{air}$) variable from GLDAS Noah Land Surface Model v2.1 (Rodell et al., 2004), MODIS 15 Leaf Area Index ($LAI$) (Myneni et al., 2021), and district-wise sugarcane crop yield from International Crops Research Institute for the Semi-Arid Tropics database.

Dwivedi, P. and Yadav, B. K.: Inference of hydrological modelling and field-based monitoring on dynamics of heavy metals in water of Hindon Basin, Journal of Hydrology: Regional Studies, 60, 102488, 2025.

Indian Council of Agricultural Research: Indian Council of Agricultural Research, Co 0238 – The Wonder Variety of Sugarcane: https://icar.org.in/en/node/4869.

International Crops Research Institute for the Semi-Arid Tropics: District Level Database (DLD): Crops dataset,http://data.icrisat.org/dld/src/crops.html.

Myneni, R., Knyazikhin, Y., and Park, T.: MODIS/Terra Leaf Area Index/FPAR 8-Day L4 Global 500m SIN Grid V061, NASA EOSDIS Land Processes Distributed Active Archive Center (DAAC) data set, MOD15A12H. 061, 2021.

Rodell, M., Houser, P., Jambor, U., Gottschalck, J., Mitchell, K., Meng, C.-J., Arsenault, K., Cosgrove, B., Radakovich, J., and Bosilovich, M.: The global land data assimilation system, Bulletin of the American Meteorological society, 85, 381-394, 2004.

---

## Author Response (AR2)

Reviewer #1:

"This manuscript presents a novel probabilistic water balance data fusion approach for calibrating multi-scale hydrological datasets. The methodology is innovative, addressing the challenge of reducing uncertainties in datasets by integrating them through water balance constraints. The approach provides a framework for both basin-scale and pixel-scale applications. The application to the Hindon River Basin demonstrates practical utility, with reasonable error estimates and clear improvements in data consistency. The paper is well-written, structured, and accessible, making a substantial contribution to water resource management and hydrological modeling. However, some areas, such as the clarity of methodological details and validation against independent data, could be strengthened to enhance the robustness and reproducibility of the findings. Suggestions are as follows:"

We would like to thank the referee for the time and effort reviewing our manuscript, and for the valuable feedback received. We are pleased that the reviewer found the paper well written and appreciate the recognition of the novelty of the presented methods. In the following, we address the reviewer's detailed comments.

Reviewer comment 1:
"In Section 2, beginning on line 117, you describe the Hindon Basin and the separation of two irrigation seasons (Kharif and Rabi), yet it is unclear if the rotated crops use the same land or if they are in adjacent regions. It would be helpful to add a sentence or two clarifying this."

Reply on comment 1:
We have added a sentence (line 121 in the clean revised version) to mention that the distributaries take off from main canals to serve fixed command areas irrigated year-round, with crops rotated between Kharif and Rabi crops.

Reviewer comment 2:
In your results, the validation could be strengthened. Are you able to compare your estimates against any in-situ records? Reported standard errors are useful, but which component dominates the uncertainty (precip, evaporation, storage, discharge, canal imports)? Standard errors are provided but there is no discussion of comparisons with independent ground-truth data or other datasets not used in calibration. Including such validation would enhance confidence in the results.

Reviewer comment 3:
Discussion would benefit from a short explanation on generalization. For example, can this approach work in snow dominated or urban catchments or is it basin specific?

Reply on comments 2 and 3:
Indeed, it will be useful to report the order of the standard errors of the different water balance variables, and it might be more interesting to highlight this at the start of the paper. Therefore, we edited the abstract to reflect the order. We also agree that it's good to reflect on model generalizability and validation; therefore, we added a separate section 6.4. We have also looked into additional data, such as weather and crop variables, to further validate our results. We added figure C4, comparing the trend in our estimated evaporation with independent crop yield data, and we discussed these results in section 3.5 (lines 431-444 in the clean revised version).

Reviewer comment 4:
"Figures with more than one panel (starting with Figure 2) need tags (a, b, c, etc) and the caption should refer to each panel specifically for clarity (like you did for Figure 4). In Table 2, indicate the meaning of the underlined values. Table 3, Table 4, again indicate the bold and underline importance."

Reply on comment 4:
Thank you for the suggestions regarding figures and tables. We have added the alphabetic labels for the subplots of Figures 2 and 3 and edited their captions. We have also edited the captions of tables 2, 3, and 4.

Reviewer #2:

This work proposes a two-step method, i.e., from basin scale to grid scale, to produce a water budget closed datasets by introducing the Bayesian model with predefined prior distribution and posterior parameter estimation, considering the covariance between water components and the entire time series, under a specific case study in the severely irrigated Hindon River Basin, India. This work tried to introduce an innovative theoretical basis and apply it to a basin with abundant discharge records. Although the logical structure of the work is clear, the theoretical introduction and the equations are hard to follow since there is no deductive process of the equations provided and the code and data are not accessible with the provided links. This makes it hard for the readers to follow the work and estimate the robustness of the work.

We would like to thank the referee for the time and effort reviewing our manuscript, and for the valuable feedback received. We understand that the theoretical introduction of the methods and equations might be hard to follow. We have improved the flow of equations by adding intermediate steps in the equations that more clearly show how one equation leads to the next (see details below). Also, we have updated the link to the data and software, as follows:

"Code and data availability:
All software used to collect the data is available at https://doi.org/10.5281/zenodo.11148992 . The source code developed for this research is available via https://zenodo.org/records/17348274."

Reviewer major comment 1:
"the spatial scale problem is not thoroughly discussed. How were the different scales between water components in the budget closure equation handled? Is the resolution 50 km really feasible in such a small basin that is only one pixel wide and two pixel height?"

Reply on major comment 1:

We agree this point deserves more attention, and we added lines 176-181 (in the clean revised version of the paper) to reflect on the challenge of the scale difference between the water balance variables.

As for the 50 km, this refers to the spatial correlation length ($l_s$) that we specify a priori to generate the covariance matrices of precipitation and evaporation. The basic assumption here is that the values of the grid cells in the spatial domains of precipitation and evaporation are spatially correlated, i.e., the correlation between the grid values depends on the distance between the grid locations. Grid cells that are close to each other tend to have high correlation. As the spatial separation between grid cells increases, the correlation decreases. Often, information on the correlation length parameter is not available. In such a case and given the differences in scales between the data sets, we rely on specifying a mid-range value of the prior correlation length, which is about half the basin's length from North to South. Finally, we assess the sensitivity of the results to the chosen correlation length in sect 6.2.

Reviewer major comment 2:
"L173/L884-885. About the gap-filling method, the authors made the assumption that the canal operations do not vary widely between years in which the conditions are similar. Is it possible to use the existing data to validate the assumption? I mean compare the data in the years with similar conditions to check whether that assumption is tenable."

Reply on major comment 2:
Unfortunately, we are constrained by the amount of data available, making cross-validation difficult. Specifically, we at most have five years of data, and these years can be distributed between wet, dry, and normal years, leaving us with no out-of-sample data for cross-validation. E.g., if among these 5 years, 2 years are dry, we simply took the average of these two years to fill all other missing dry years. In cases where we know that a distributary might significantly contribute to the overall imports to the basin, e.g., the Deoband branch, we follow a different approach that avoids any average-based related biases and relies on using the design discharge and operation time to estimate the canal water imports for the missing years. In this vein, using the canal design capacities for gap-filling results in conservative (upper

bound) estimates of the irrigation water imports, which can also be seen as acceptable approximate initial values for the Bayesian methods used in the study. Additionally, in sect 6, we include scenario 2, where, instead of filling the missing years with the data average of years with similar conditions, we use the upper bound estimates (design capacities multiplied by the operation time). The results show insensitivity to the canal water estimates used, as these are much smaller than the surface exchanges (e.g., precipitation and evaporation). To account for potential uncertainties around all assumptions we make to generate the corresponding full timeseries, we attach a large prior uncertainty (25%) to these gap-filled estimates, i.e., a wide confidence interval around their values.

Reviewer  major comment 3:
"It will be much clearer and easier to follow if a framework diagram is provided in the method section."

Reply on major comment 3:
Good suggestion, a flow diagram can aid the reader in grasping the overall flow of the methods section. We added figure 2.

Reviewer  major comment 4:
"About the matrix variables mentioned in all equations, for example, in Eq. 5, it is better to provide the size parameters of each matrix."

Reviewer  major comment 5:
"For me, the relationships between equations are quite independent and the connections are weak. For example, Eq. 3-8, the input and output of each equation are vague thus hard to understand the method itself as a whole. The same issue exists for the entire theoretical part."

Reply on major comments 4 and 5:
We added the information about the size of the evaporation and precipitation correlation matrices. These are now mentioned in the following subsections, which have been edited to show additional intermediate steps when going from one equation to the next. These subsections now read as follows:

[revised manuscript text omitted]

Reviewer  major comment 6:
"L545. The labels and tick marks of x and y missed in Fig. 8".

Reply on major comment 6:
Since the river discharge data are classified (can't be made publicly available), the labels and tick marks of x and y are not shown in Fig. 8. We have updated the caption of Fig. 8 to mention this.

Reviewer  minor comment 1:
"L125 the abbreviation of Central Water Commission (CWC) should be explained near the figure instead later in L168".

Reply on minor comment 1:
That's true; we have defined CWC in the Figure 1 caption.

Reviewer minor comment 2:
"L241. the symbols mpt and vpt with Eq.7-8 are different."

Reply on minor comment 2:
We have removed these notations for conciseness and to enhance clarity.

Reviewer minor comment 3:
"L255 I think "Evapotranspiration" is better than "Evaporation" throughout the paper."

Reply on minor comment 3:
At the outset of introducing the probabilistic water balance model (line 131 in the revised version of the manuscript), we define evapotranspiration as evaporation (including transpiration). Later, we use the term "evaporation" throughout the paper for conciseness.

Reviewer minor comment 4:
"L581. There is no ground-water level data? It is weird that the discharge of the canals have been paid great attention while no ground-water data available in such a heavily ground-water-based irrigated basin."

Reply on minor comment 4:
The reviewer is right, the inclusion of groundwater level as an independent evaluation would be valuable; however, in this study, we only considered the total water storage. In a follow-up study, we plan to extend the methodology presented here to incorporate detailed information, such as groundwater pumping, soil moisture, and groundwater level data, where we also focus on separating the rootzone from the groundwater contribution. See also our response to Reviewer 1's comment on validation, and the newly added section 6.4, reflecting on this.

Abramowitz, M. and Stegun, I. A.: Handbook of mathematical functions with formulas, graphs, and mathematical tables, US Government printing office1968.

Chen, H., Ding, L., and Tuo, R.: Kernel packet: An exact and scalable algorithm for Gaussian process regression with Matérn correlations, Journal of machine learning research, 23, 1-32, 2022.

Handcock, M. S. and Stein, M. L.: A Bayesian analysis of kriging, Technometrics, 35, 403-410, 1993.

Hensman, J., Matthews, A., and Ghahramani, Z.: Scalable variational Gaussian process classification, Artificial intelligence and statistics, 351-360,

Schoups, G. and Nasseri, M.: GRACEfully closing the water balance: A data-driven probabilistic approach applied to river basins in Iran, Water Resources Research, 57, e2020WR029071, 2021.

Sun, Y., Bowman, K. P., Genton, M. G., and Tokay, A.: A Matérn model of the spatial covariance structure of point rain rates, Stochastic Environmental Research and Risk Assessment, 29, 411-416, 2015.